# Most Influential Subset Selection: Challenges, Promises, and Beyond

**Yuzheng Hu**[1]   **Pingbang Hu**[2]   **Han Zhao**[1]   **Jiaqi W. Ma**[2]
[1]Department of Computer Science   [2]School of Information Sciences
University of Illinois Urbana-Champaign
{yh46,pbb,hanzhao,jiaqima}@illinois.edu

## Abstract

How can we attribute the behaviors of machine learning models to their training data? While the classic influence function sheds light on the impact of individual samples, it often fails to capture the more complex and pronounced collective influence of a set of samples. To tackle this challenge, we study the Most Influential Subset Selection (MISS) problem, which aims to identify a subset of training samples with the greatest collective influence. We conduct a comprehensive analysis of the prevailing approaches in MISS, elucidating their strengths and weaknesses. Our findings reveal that influence-based greedy heuristics, a dominant class of algorithms in MISS, can provably fail even in linear regression. We delineate the failure modes, including the errors of influence function and the non-additive structure of the collective influence. Conversely, we demonstrate that an adaptive version of these heuristics which applies them iteratively, can effectively capture the interactions among samples and thus partially address the issues. Experiments on real-world datasets corroborate these theoretical findings and further demonstrate that the merit of adaptivity can extend to more complex scenarios such as classification tasks and non-linear neural networks. We conclude our analysis by emphasizing the inherent trade-off between performance and computational efficiency, questioning the use of additive metrics such as the Linear Datamodeling Score, and offering a range of discussions.

## 1   Introduction

Unraveling the intricate connections between data and model predictions is critical in machine learning, particularly in high-stakes decision-making contexts such as healthcare, economics, and public policy [Bracke et al., 2019, Rudin, 2019, Amarasinghe et al., 2023]. A better understanding of these connections allows tackling tasks like data cleaning [Teso et al., 2021], model debugging [Guo et al., 2021], and assessing the robustness of inferential results [Broderick et al., 2020], all key to enhancing model interpretability and fostering trust between machine learning practitioners and domain experts. Among the various methodologies, the influence function adopted by Koh and Liang [2017] stands out as a particularly effective tool, sparking extensive research into identifying influential individual samples [Barshan et al., 2020, Schioppa et al., 2022, Grosse et al., 2023].

Nevertheless, focusing solely on the influence of individual samples is often insufficient. In many scenarios, it is necessary to understand how sets of samples jointly affect model predictions. These include uncovering biases associated with specific demographic groups [Chen et al., 2018], fairly allocating credits among crowdworkers [Arrieta-Ibarra et al., 2018], and detecting trends and signals that emerge collectively within the data [Yang et al., 2020]. Gaining such insights is crucial for a more comprehensive understanding of model behaviors.

38th Conference on Neural Information Processing Systems (NeurIPS 2024).

In pursuit of advancing this field, in this paper, we delve into the most influential subset selection (MISS) problem [Fisher et al., 2023]. MISS attempts to find a set of samples that, when removed from the training set, results in the most significant change of a pre-defined target function. In essence, it measures the *worst-case* collective influence.

**Contributions.** We provide a comprehensive analysis of existing algorithms to tackle MISS, revealing their weaknesses and strengths, and discussing the challenges and important considerations for future research. To summarize our contributions:

- We systematically study the failure modes of *influence-based greedy heuristics*, a dominant class of algorithms in MISS that assign a static score to each sample and subsequently perform a greedy selection. Specifically, the error of influence function, as well as the inability to incorporate the non-additive structure of the collective influence, can cause these heuristics to fail in MISS even in simple linear regression.

- In contrast, we demonstrate the effectiveness of the *adaptive greedy algorithm* that dynamically updates the score for each remaining sample in response to selections already made. The improvement mainly comes from its ability to capture the nuanced interactions among samples.

- We conduct experiments on both synthetic and real-world datasets. The experimental results not only corroborate the theoretical findings but also extend to more complex settings including classification tasks and non-linear models, showcasing the consistent benefits of adaptivity.

- We discuss the inherent trade-offs between performance and efficiency in MISS, and the potential drawbacks of additive metrics such as Linear Datamodeling Score, among others.

**Concurrent work.** We acknowledge a concurrent work [Huang et al., 2024], which was posted around the same time as ours. Huang et al. [2024] investigate the Maximum Influence Perturbation problem [Broderick et al., 2020], which is equivalent to MISS. Both studies analyze the additive assumption and the adaptive greedy algorithm in OLS, but they differ in the theoretical results. Notably, we formally prove the failure of LAGS in solving MISS under a specific data generation process, uncovering the phenomena of amplification and cancellation. Huang et al. [2024] analyze the approximation error of variants of LAGS by comparing the closed-form expression of the approximate algorithm and the actual effect.

## 2 Preliminaries

### 2.1 Problem statement

Consider a prediction task (e.g., regression or classification) with an input space $\mathcal{X} \subset \mathbb{R}^d$ and a target space $\mathcal{Y} \subset \mathbb{R}$. The prediction task aims to learn a function $f(\theta, \cdot) : \mathcal{X} \to \mathcal{Y}$ parameterized by $\theta \in \mathbb{R}^q$. Specifically, denote $\{(x_i, y_i)\}_{i=1}^n$ as the training samples and $L(\cdot, \cdot)$ as the loss function (e.g., squared error or cross-entropy), we aim to solve the following optimization problem:

$$\hat{\theta} = \underset{\theta \in \mathbb{R}^q}{\arg\min} \frac{1}{n} \sum_{i=1}^n L(f(\theta, x_i), y_i). \tag{1}$$

A key notion for analyzing the influential samples is the optimal model parameters after removing a subset of training samples. Denote $[n] = \{1, 2, \cdots, n\}$ and the set of indices as $S \subset [n]$, this corresponds to

$$\hat{\theta}_{-S} = \underset{\theta \in \mathbb{R}^q}{\arg\min} \frac{1}{n} \sum_{i \notin S} L(f(\theta, x_i), y_i). \tag{2}$$

Note that we do not adjust the normalizing constant as it does not affect the optimal solution to Eq.(2). Finally, denote $\phi : \mathbb{R}^q \to \mathbb{R}$ as the *target function*, which takes the model parameters as input and returns a quantity of interest (e.g., the prediction on a test sample or the sign of its first coefficient). We now formally define the most influential subset selection problem.

**Definition 2.1** (Most Influential Subset Selection (MISS)). *Given a positive integer $k \ll n$, the $k$-Most Influential Subset Selection ($k$-MISS) problem refers to this discrete optimization problem:*

$$S_{\mathrm{opt},k} = \underset{S \subset [n], |S| \le k}{\arg\max} A_{-S}, \text{ where } A_{-S} := \phi(\hat{\theta}_{-S}) - \phi(\hat{\theta}). \tag{3}$$

We refer to $A_{-S}$ as the *actual effect* of removing $S$. For clarity, we refer to the actual effect as the *individual effect* when $|S| = 1$ and the *group effect* otherwise. Essentially, MISS aims to identify a subset with bounded size, such that its removal from the training samples will lead to the maximum actual effect. It can be viewed as analogous to adversarial examples [Biggio et al., 2013, Szegedy et al., 2014], in that both characterize the alteration of model behaviors in the *worst case*, but MISS operates on the training data space and during training time.

Unfortunately, the naive approach of enumerating all possible subsets has an exponential time complexity in $k$, rendering it computationally intractable in practice. In fact, even in the context of linear regression, a variant of MISS (where the target function depends on $S$) known as *robust regression* [Andersen, 2007] is proved to be NP-hard [Price et al., 2022]. To tackle this challenge, researchers have proposed various greedy heuristics to select an *approximately* most influential subset.

## 2.2 Influence-based greedy heuristics

One of the most prominent algorithms for MISS, ZAMinfluence, was introduced by Broderick et al. [2020] and applied to assess the robustness of inferential results in earlier econometric studies [Attanasio et al., 2015, Angelucci et al., 2015]. It builds upon the classic influence function [Koh and Liang, 2017] from robust statistics literature [Hampel, 1974, Hampel et al., 2005], extending its application from individual samples to a set of samples. A similar approach has been employed by Koh et al. [2019] to estimate group effects. We defer a detailed review of the literature to Section 7.

**Definition 2.2** (Upweighted objective). *We denote the optimal solution to the upweighted objective w.r.t. a set of indices $S$ as*

$$\hat{\theta}_{-S}(\delta) := \arg\min_{\theta \in \mathbb{R}^q} \frac{1}{n} \sum_{i=1}^{n} L(f(\theta, x_i), y_i) + \delta \sum_{i \in S} L(f(\theta, x_i), y_i). \tag{4}$$

It is straightforward to see that $\delta = 0$ corresponds to $\hat{\theta}$, while $\delta = -\frac{1}{n}$ corresponds to $\hat{\theta}_{-S}$. Similar to the influence function of individual samples [Koh and Liang, 2017], the influence of a set $S$ can be characterized by the local perturbation of $\hat{\theta}_{-S}(\delta)$ around $\delta = 0$. This quantity is well-defined when $L$ is strictly convex and can be computed via the Implicit Function Theorem [Krantz and Parks, 2002].

**Definition 2.3** (Influence function of a set). *The influence of upweighting $S$ on the parameters is:*

$$\mathcal{I}(S) := \frac{d\hat{\theta}_{-S}(\delta)}{d\delta}\bigg|_{\delta=0} = -H_{\hat{\theta}}^{-1} \sum_{i \in S} \nabla_\theta L(f(\hat{\theta}, x_i), y_i), \tag{5}$$

*where $H_{\hat{\theta}} = \frac{1}{n} \sum_{i=1}^{n} \nabla_\theta^2 L(f(\hat{\theta}, x_i), y_i)$ is the Hessian of the loss function at $\hat{\theta}$.*

Using the chain rule and note that $\hat{\theta}_{-S} = \hat{\theta}_{-S}(-\frac{1}{n})$, the actual effect can be estimated via the first-order approximation:

$$A_{-S} \approx -\frac{1}{n} \cdot \frac{d\phi(\hat{\theta}_{-S}(\delta))}{d\delta}\bigg|_{\delta=0} = \frac{1}{n} \nabla_\theta \phi(\hat{\theta})^\top H_{\hat{\theta}}^{-1} \sum_{i \in S} \nabla_\theta L(f(\hat{\theta}, x_i), y_i). \tag{6}$$

The key observation is that the right-hand side of Eq. (6) displays an *additive* structure so that the group effect can be approximated by a summation of individual influences. This naturally yields the ZAMinfluence algorithm, which involves 1) calculating $v_i = \nabla_\theta \phi(\hat{\theta})^\top H_{\hat{\theta}}^{-1} \nabla_\theta L(f(\hat{\theta}, x_i), y_i)$ for each $i \in [n]$; 2) sorting $v_i$'s; 3) returning the top $i$'s with positive $v_i$. In fact, a series of studies in MISS [Wang et al., 2023, Yang et al., 2023a, Chhabra et al., 2024] follow a similar approach: they score individual samples using variants of influence functions, and then greedily select those with the highest positive scores. We refer to these algorithms as *influence-based greedy heuristics*.

These heuristics are powerful in two aspects. The first is their broad applicability: they can be applied to *any* $Z$-estimator of a twice-differentiable objective function [Broderick et al., 2020] to obtain an influential subset w.r.t. *any* differentiable target function. The second is their computational efficiency: once we have computed the scores for each sample, they can be executed in linear to log-linear time complexity. However, a major drawback of these heuristics is the lack of *provable* guarantees. It is well-known that even the influence estimates of individual samples can be fragile and erroneous,

especially in complex models like neural networks [Basu et al., 2021, Bae et al., 2022]. A more significant concern lies in the additivity assumption implicitly adopted by these heuristics (also see Guu et al. [2023] for discussions), as it fails to account for the interactions among samples. We critically examine these issues in Section 3.

# 3 Pitfalls of greedy heuristics in Most Influential Subset Selection

In this section, we delve into the influence-based greedy heuristics introduced in Section 2, providing a comprehensive study of their limitations in solving MISS within the context of linear regression.

**Setup and notation.** In standard linear regression, each $x_i \in \mathbb{R}^d$ represents a vector of covariates, and $y_i$ stands for a real-valued label. The first coordinate of each $x_i$ is set to 1 to account for the intercept term. We stack the row vectors $x_i^\top$ to form the design matrix $X \in \mathbb{R}^{n \times d}$ and concatenate the $y_i$'s into the target vector $y \in \mathbb{R}^n$. We assume the labels are generated as follows: there exists a $\theta^* \in \mathbb{R}^d$ (note $q = d$), a noise parameter $\varepsilon > 0$ and some $p$, such that

$$e = (\varepsilon, 0, \cdots, 0, p\varepsilon)^\top \in \mathbb{R}^n, \quad y = X\theta^* - e. \tag{7}$$

For a subset $S$, $X_S$ and $y_S$ denote the corresponding covariates and responses, while $X_{-S}$ and $y_{-S}$ represent their complements. To ensure the uniqueness of the optimal solution, we assume $N = X^\top X$ is invertible, and that $\sum_{i=2}^{n-1} x_i x_i^\top$ is also invertible (when this assumption is violated, our results naturally extend to ridge regression). The hat matrix is denoted as $H = X N^{-1} X^\top$. The diagonal element $h_{ii}$ of $H$ represents the *leverage score* of $x_i$, and the off-diagonal element $h_{ij}$ represents the *cross-leverage score* [Chatterjee and Hadi, 2009] between $x_i$ and $x_j$. The Ordinary Least Squares (OLS) estimator is given by

$$\hat{\theta} = \arg\min_\theta \frac{1}{n}\|X\theta - y\|^2 = N^{-1}\sum_{i=1}^n x_i y_i. \tag{8}$$

Let $\hat{y}_i = x_i^\top \hat{\theta}$ be the prediction and $r_i = \hat{y}_i - y_i$ be the negative residual for the $i$-th sample. Throughout Sections 3 and 4, we focus on the linear target function $\phi(\theta) = x_{\text{test}}^\top \theta$ for $x_{\text{test}} = \frac{x_1 + p x_n}{p+1}$, whose first coordinate is also 1. This choice of $x_{\text{test}}$ is intentional: it greatly simplifies the analysis by making most of the individual effects negative, as reflected in Figures 1 to 3 and the calculations in Appendix A.1. Furthermore, due to the continuous nature of the problem, our conclusions hold for a set of $x_{\text{test}}$ with non-zero Lebesgue measure.

## 3.1 Influence function is not accurate (even) in linear models

Influence function is widely acknowledged as an accurate alternative of leave-one-out re-training in linear models [Koh and Liang, 2017, Basu et al., 2021, Bae et al., 2022]. In this section, however, we challenge this viewpoint by pointing out a previously overlooked fact: the influence function fails to incorporate the leverage scores of individual samples in linear regression, which could result in its failure in selecting the most influential sample (i.e., 1-MISS).

Plugging the squared loss into Eq. (5), we have $\mathcal{I}(S) = -nN^{-1}\sum_{i \in S} x_i r_i$. Therefore, ZAMinfluence assigns $v_i = x_{\text{test}}^\top N^{-1} x_i r_i$ to each sample. We refer to them as *influence estimates*. On the other hand, it is well-known in the statistics literature [Beckman and Trussell, 1974, Cook, 1977] that

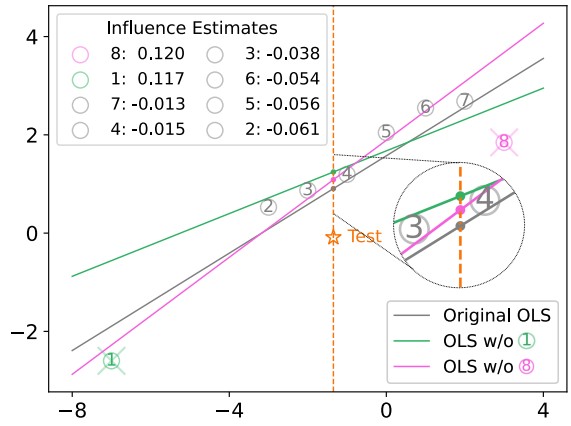

Figure 1: Influence estimates suffer from disparate levels of under-estimation, leading to the failure of 1-MISS

$$\hat{\theta}_{-\{i\}} - \hat{\theta} = \frac{N^{-1} x_i r_i}{1 - h_{ii}}. \tag{9}$$

Consequently, the change in the target function is given by $A_{-\{i\}} = \frac{x_{\text{test}}^\top N^{-1} x_i r_i}{1 - h_{ii}}$, which deviates from the influence estimate by a factor of $1/(1 - h_{ii})$ and implies under-estimation (a phenomenon which was also reported in Koh et al. [2019]). This is particularly concerning when a sample has a high leverage score (e.g., an outlier [Chatterjee and Hadi, 1986]): in this case, the influence function substantially under-estimates the individual effect, potentially leading to the failure of 1-MISS. We illustrate this intuition in Figure 1: while point ⑧ is scored highest by the influence function, it is however removing point ① (which has the highest leverage score) that leads to the greatest change in the prediction on the test sample. More generally, we present the following theorem illustrating the failure of ZAMinfluence in 1-MISS, with the proof detailed in Appendix A.2.

**Theorem 3.1.** *Assume $h_{11} > h_{nn}$. Under the label generation process described in Eq.(7), there exists some $p$, such that ZAMinfluence fails to select the most influential sample.*

> **Takeaway:** Even when the influence estimates have high *correlation* with the individual effects, they can be misleading for extreme samples. As a result, the influence function may not be a reliable tool for MISS.

### 3.2 Violation of the additivity assumption: amplification and cancellation

Note that the individual effects $A_{-\{i\}}$'s can be computed efficiently for linear regression (this is generally infeasible for more complicated tasks) by correcting the influence estimates $v_i$'s with their corresponding leverage scores. Hence, a natural alternative is to directly perform greedy selection based on the $A_{-\{i\}}$'s. We refer to this method as *Leverage-Adjusted Greedy Selection* (LAGS). Nevertheless, we will illustrate in this section that even with perfect individual influence estimation, LAGS may still fall short in MISS due to violations of the additivity assumption.

We start by computing the closed-form of $A_{-S}$. The proof can be found in Appendix A.3.

**Proposition 3.2.** *For any set of indices $S$, we have*

$$A_{-S} := \phi(\hat{\theta}_{-S}) - \phi(\hat{\theta}) = x_{\text{test}}^\top N^{-1} X_S^\top \left( I_k - X_S N^{-1} X_S^\top \right)^{-1} (X_S \hat{\theta} - y_S). \tag{10}$$

**Remark 3.3.** *Denote $M_S = X_S N^{-1} X_S^\top$. It is straightforward to see that replacing the Neumann series $(I_k - M_S)^{-1} = I_k + M_S + M_S^2 + \cdots$ by the identity matrix yields the influence estimates, i.e., the first-order approximation. We further prove in Appendix A.4 that there is a one-to-one correspondence between the Taylor series of $\hat{\theta}_{-S}(\delta)$ and the Neumann series: for any $k \in \mathbb{N}^+$, the $k$-th order approximation of $\hat{\theta}_{-S}(\delta)$ is equivalent to truncating the Neumann series at $M_S^{k-1}$. On the other hand, LAGS is based on the diagonal approximation of $(I_k - M_S)$.*

To systematically study the failure mode of LAGS, we consider $S = \{i, j\}$. In this case,

$$A_{-\{i,j\}} = x_{\text{test}}^\top \left( \frac{(1 - h_{jj})N^{-1}x_i r_i + (1 - h_{ii})N^{-1}x_j r_j + h_{ij}N^{-1}(x_i r_j + x_j r_i)}{(1 - h_{ii})(1 - h_{jj}) - h_{ij}^2} \right)$$

$$= \frac{(1 - h_{ii})(1 - h_{jj})(A_{-\{i\}} + A_{-\{j\}}) + h_{ij}x_{\text{test}}^\top N^{-1}(x_i r_j + x_j r_i)}{(1 - h_{ii})(1 - h_{jj}) - h_{ij}^2}. \tag{11}$$

From Eq.(11), we identify two primary factors contributing to the non-additivity of the group effect: the cross-leverage score $h_{ij}$ in the denominator, which can lead to *super-additivity* by inflating the sum of individual effects, and the cross terms $x_{\text{test}}^\top N^{-1}(x_i r_j + x_j r_i)$ in the numerator, which may result in *sub-additivity* through the neutralization of individual effects. We refer to these phenomena as "amplification" and "cancellation," respectively, and will delve into how they provably lead to the failure of LAGS in what follows.

**Amplification.** Amplification occurs when the group effect of a set substantially exceeds the sum of individual effects. As suggested by Eq.(11), this phenomenon is pronounced when the cross-leverage score is high. Therefore, we focus on scenarios where there are $c \geq 2$ identical copies of a sample, in which case the cross-leverage score becomes the leverage score. Intuitively, this setting can be generalized to a cluster of similar samples. We first prove a useful result in this context.

**Proposition 3.4.** *Suppose there are $c$ copies of $(x_i, y_i)$. We have*

$$\frac{A_{-\{i\}^c}}{A_{-\{i\}}} = \frac{c \cdot (1 - h_{ii})}{1 - ch_{ii}} > c, \tag{12}$$

*where $A_{-\{i\}^c}$ denotes the group effect of removing all $c$ copies of $(x_i, y_i)$.*

The proof can be found in Appendix A.5. It suggests that the group effect not only surpasses the sum of individual effects, but their ratio can be unbounded as $h_{ii} \to \frac{1}{c}$. Put differently, a sample with minor influence can collectively cause a substantial effect when grouped with similar ones. In MISS, this could lead to the failure of LAGS when there is a cluster of samples with high leverage scores yet do not have the largest individual effects. This intuition is illustrated in Figure 2: while points ⑦ and ⑧ (the pink cluster) have the highest individual effects due to their large residuals, points ① and ② (the green cluster) with high leverage scores constitute the most influential size-2 subset.

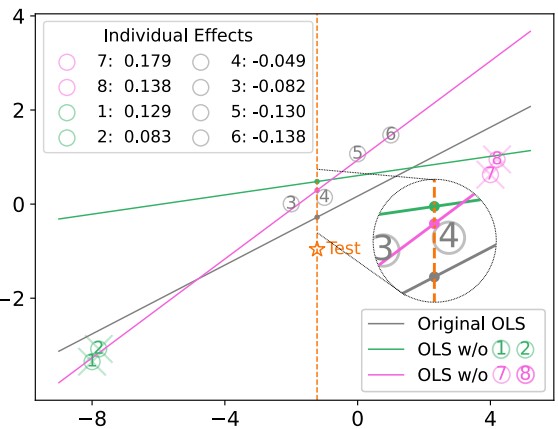

Figure 2: LAGS fails in 2-MISS due to amplification

We show a generalization of this example in the following theorem and defer its proof to Appendix A.6.

**Theorem 3.5.** *Suppose there are $c$ copies of $(x_1, y_1)$ and $(x_n, y_n)$, and that $h_{11} > h_{nn}$. Under the label generation process described in Eq.(7), there exists some $p$, such that LAGS fails in $c$-MISS.*

**Cancellation.** Cancellation happens when the group effect of a set $S$ is less than one of its subsets $S'$, indicating that removing $S \setminus S'$ induces a negative effect.

In this case, cancellation is equivalent to $A_{-\{1,n\}} < A_{-\{n\}}$ (we assume w.l.o.g. that $A_{-\{n\}} > A_{-\{1\}}$). From Eq.(11), this inequality is likely to hold when $A_{-\{1\}}$ has a small magnitude compared to $A_{-\{n\}}$, and the sign of $h_{1n}$ differs from that of $\frac{r_n}{r_1}$. If we further have that $A_{-\{1\}}$ and $A_{-\{n\}}$ are the top-2 positive individual effects (which guarantees that they will be selected by the greedy algorithm), then LAGS will fail in this context.

We illustrate this in Figure 3: although points ⑧ and ① have the top-2 individual effects and are positive, their group effect as a size-2 subset is less than the individual effect of point ⑧.

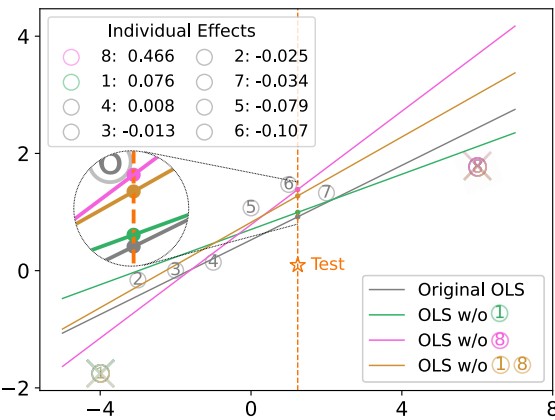

Figure 3: LAGS fails in 2-MISS due to cancellation

We present a more general result in the following theorem and defer its proof to Appendix A.7.

**Theorem 3.6.** *Assume $h_{1n} \neq 0$. Under the label generation process described in Eq.(7), there exists some $p$, such that LAGS fails in 2-MISS.*

---

**Takeaway:** LAGS provably works for MISS when all cross-leverage scores are zero, but can fail with even a single non-zero cross-leverage score. This highlights the algorithm's fragility.

---

## 4 Promises of the adaptive greedy algorithm

Given the limitations of LAGS, a pertinent question arises: is it possible to capture the non-additive structure of the joint effect without enumerating subsets? In this section, we examine a refined heuristic proposed by Kuschnig et al. [2021], and provide a theoretical analysis following our framework in Section 3. Kuschnig et al. [2021] originally introduced this refined algorithm in the

context of linear regression, which applies to general influence-based greedy heuristics. The idea is to *adaptively* build the influential subset. Specifically, the algorithm works by 1) refitting the model on the current dataset and recalculating the individual effect or influence estimate for each sample; 2) excluding the most influential sample from the current dataset; 3) adding it to the influential subset. This iterative process is repeated until the subset reaches the desired size. We refer to this as the *adaptive greedy algorithm*.

It is empirically observed that the adaptive greedy algorithm outperforms LAGS in linear regression [Kuschnig et al., 2021]. In this section, we further aim to provide theoretical support for the benefits of *adaptivity*. Specifically, we will show that in scenarios where LAGS fail due to cancellation, the adaptive greedy algorithm can effectively address this problem by leveraging a scoring function that captures the marginal contributions relative to the removal of the most influential sample.

Following the cancellation setup, $(x_n, y_n)$ is the most influential sample w.r.t. the full dataset. We denote $A'_{-\{i\}}$ as the actual effect of removing $(x_i, y_i)$ for $1 \leq i \leq n-1$ *after* the removal of $(x_n, y_n)$. Essentially, $A'$ is the scoring function employed in the second step of the adaptive greedy algorithm. We start by proving two useful properties of $A'$ (the proof is deferred to Appendix B.2).

**Proposition 4.1.** *The scoring function $A'$ satisfies the following properties:*

1. ***Sign consistency**: $A'_{-\{i\}}$ and $(A_{-\{i,n\}} - A_{-\{i\}})$ have the same sign for $1 \leq i \leq n-1$;*
2. ***Order preservation**: $\{A'_{-\{i\}}\}_{i=2}^{n-1}$ and $\{A_{-\{i,n\}}\}_{i=2}^{n-1}$ are order-isomorphic.*

These properties have significant implications. The first property indicates that $A'$ is a more reliable scoring function as it captures the marginal contribution of each sample *relative to the removal of* $(x_n, y_n)$. Hence, in the cancellation setup, $A'$ will not choose $(x_1, y_1)$, even though $A_{-\{1\}}$ represents the second-largest individual effect and is positive. In contrast, the actual effect $A$, which reflects the marginal contribution of each sample relative to the full dataset, does not account for how a newly selected sample interacts with those already selected. The second property further guarantees the success of MISS based on $A'$. Formally, we prove the following for the adaptive greedy algorithm.

**Theorem 4.2.** *Under the label generation process described in Eq. (7), suppose $A_{-\{1\}}, A_{-\{n\}} > 0$, $A_{-\{1,n\}} < A_{-\{n\}}$ (indicating cancellation), and that $n \in S_{\text{opt},2}$ (i.e., $(x_n, y_n)$ is contained in the most influential subset), then the adaptive greedy algorithm solves 2-MISS.*

*Proof.* We first show that the condition $A_{-\{1,n\}} < A_{-\{n\}}$ implies that $(x_n, y_n)$ is the most influential sample (the proof is deferred to Appendix B.3). This ensures that the adaptive greedy algorithm will select $(x_n, y_n)$ in the first step. We now discuss two cases separately.

**Case 1:** If $A_{-\{i,n\}} - A_{-\{n\}} < 0$ for every $2 \leq i \leq n-1$, then $S_{\text{opt},2} = \{n\}$. Furthermore, by the first property of Proposition 4.1 we have $A'_{-\{i\}} < 0$ for $1 \leq i \leq n-1$. This implies that the adaptive algorithm will return $\varnothing$ in the second step since no scores are positive, as desired.

**Case 2:** If there exists some $2 \leq i \leq n-1$, such that $A_{-\{i,n\}} - A_{-\{n\}} > 0$. We denote the most influential subset as $S_{\text{opt},2} = \{i^*, n\}$. Since $A_{-\{i^*,n\}} - A_{-\{n\}} > 0$, the first property of Proposition 4.1 implies $A'_{-\{i^*\}} > 0$. Furthermore, by the second property of Proposition 4.1, the adaptive greedy algorithm will return the correct index $i^*$ in the second step.

Combining the above two cases finishes the proof of Theorem 4.2. □

**Remark 4.3.** *In the cancellation setup, our theoretical results are restricted to 2-MISS. We identify two challenges: 1) Conceptually, it is not immediately clear how to define cancellation for more than two samples; 2) Technically, proving the success of MISS is much harder than constructing a counterexample since it requires enumerating all possible subsets, whose number grows exponentially with $k$. We leave this as future work.*

---

**Takeaway:** In essence, the critical limitation of LAGS and other influence-based greedy heuristics is their reliance on a *one-pass* procedure that measures the contribution of each sample *solely in relation to the full training set*. On the other hand, the adaptive greedy algorithm considers more complex interactions between samples, akin to those in data Shapley [Ghorbani and Zou, 2019], leading to more effective subset selection.

---

# 5 Experiments

In this section, we empirically evaluate the efficacy of the adaptive greedy algorithm on real-world datasets by comparing the performance of the vanilla greedy algorithm *versus* the adaptive greedy algorithm across a range of $k$'s.[1] We cover the simple linear regression studied in Sections 3 and 4 as well as more complicated scenarios (including the classification task and non-linear neural networks) as a complement. Additional experiments on synthetic datasets can be found in Appendix C.1.

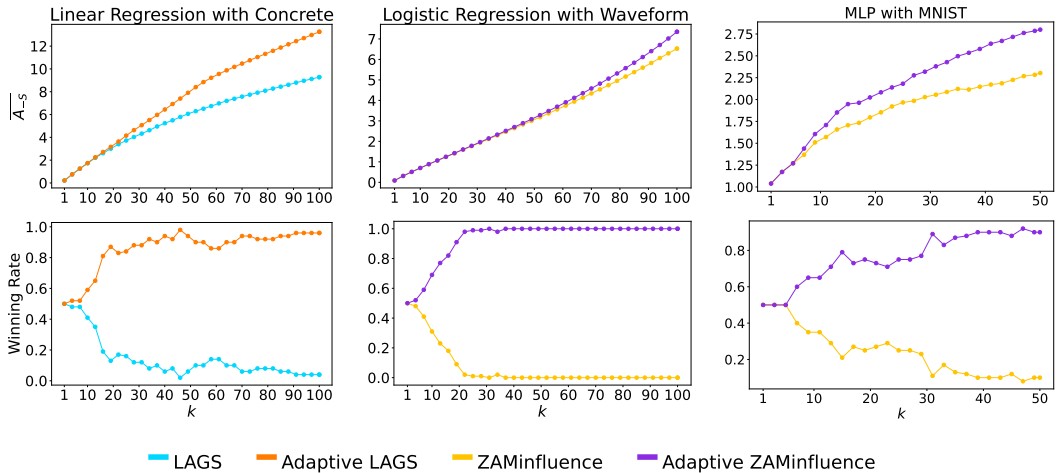

Figure 4: Adaptive Greedy v.s. Greedy Algorithm. **Row 1**: Averaged actual effect $\overline{A_{-S}}$ measures the averaged actual effect induced by the greedy and adaptive greedy algorithms. **Row 2**: Winning rate indicates the proportion of instances where one algorithm outperforms the other.

**Evaluation metrics.** We evaluate the algorithms using two metrics, the *averaged actual effect* and the *winning rate*. Given a held-out test set, we define the averaged actual effect $\overline{A_{-S}}$ as the mean of the actual effects w.r.t. each test point. A higher score of $\overline{A_{-S}}$ indicates a more influential subset is selected on average. Additionally, we report the *winning rate* across test data points in a held-out test set, namely the ratio of the algorithm outperforms the other one in terms of the actual effect $A_{-S}$.

**Target functions and greedy algorithms.** We consider two types of tasks: regression and classification. For the regression task, we adopt the target function $\phi(\theta) = x_{\text{test}}^\top \theta$ on a given test point $z := (x_{\text{test}}, y_{\text{test}})$. We utilize LAGS as the vanilla greedy algorithm. For the classification task, we consider the target function $\phi(\theta) = \log(p(z; \theta)/(1 - p(z; \theta)))$, where $p(z; \theta)$ represents the softmax probability assigned to the correct class. We opt for the ZAMinfluence as the vanilla greedy algorithm.

**Experimental setup.** For regression, we choose a popular UCI dataset *Concrete Compressive Strength* [Yeh, 2007]. For classification, we experiment with a moderate-scale UCI tabular dataset *Waveform Database Generator* [Breiman and Stone, 1988] and an image dataset MNIST [LeCun et al., 1998]. We apply logistic regression on the former and a simple 2-layer multi-layer perceptron (MLP) on the latter. We defer details of the datasets, train/test split, and MLP training to Appendix C.

**Approximated actual effect.** We address one unique challenge for the MLP: for neural networks, it is impossible to obtain the actual effect since the optimal model is not unique in general. To address this, we adopt an ensemble technique used in recent literature [Park et al., 2023]: averaging the target function's values from several independently trained models. Specifically, we train 5 models with the same initialization but different seeds. This works for both the greedy algorithm and evaluation: for the former, we estimate each model's influence with the ZAMinfluence algorithm and select the most influential subset based on the averaged influence; for the latter, we approximate the actual effect of a subset $S$ by the averaged difference of the target values of each model, trained with or without $S$.

While ensemble solves the non-uniqueness problem, it induces a significant computational burden. Noticeably, the adaptive greedy algorithm now requires retraining for ($k \times$ number of ensembles) times. To mitigate it, we use an efficient approximate variant of the ZAMifluence estimation algorithm in our implementation and devise two strategies. We defer the concrete descriptions to Appendix C.4.

---

[1]Our code is publicly available at `https://github.com/sleepymalc/MISS`.

**Results.** We present the main results in Figure 4. First, we see that as $k$ increases, the averaged actual effect $\overline{A_{-S}}$ given by both the vanilla and the adaptive greedy algorithms increase, which aligns with the intuition that removing a larger set $S$ induces a greater joint effect $\overline{A_{-S}}$. Furthermore, the adaptive greedy algorithm surpasses its vanilla counterpart across all scenarios and all $k$'s under both metrics. This implies that the benefits of adaptivity extend beyond linear regression and apply to more complicated scenarios like classification tasks and even non-linear neural networks.

Finally, for the experiment on MLP specifically, we report results of multiple random seeds in Appendix C.5 to account for the randomness in model training. The consistent results across different seeds demonstrate the robustness of the aforementioned conclusions.

## 6 Discussion

**Failure of the adaptive greedy algorithm.** While Theorem 4.2 demonstrates the advantages of the adaptive greedy algorithm, it is still not perfect. Specifically, the assumption $n \in S_{\text{opt},2}$ in Theorem 4.2 is actually necessary: if the most influential sample is not part of the most influential subset, the algorithm will make an error in the first step and cannot correct this mistake in subsequent procedures. For instance, under the amplification setup as in Theorem 3.5, it is straightforward to see that the adaptive greedy algorithms provably fail in $c$-MISS since it selects $(x_n, y_n)$ in the first place.

**Second-order approximation.** To more effectively capture the amplification effect caused by clusters of similar samples, it is essential to utilize algorithms that can detect higher-order interactions. In this context, the second-order group influence introduced by Basu et al. [2020] is a more powerful alternative. It is calculated based on the second-order approximation as described in Remark 3.3:

$$Q_{-S} = x_{\text{test}}^\top N^{-1} X_S^\top \left( I_k + X_S N^{-1} X_S^\top \right) (X_S \hat{\theta} - y_S). \tag{13}$$

From here, the original MISS can be cast as a quadratic optimization problem (see Appendix D.1) and solved via $L_1$ relaxation and projected gradient descent. Furthermore, we have $Q_{-\{1\}^c} = c^2 v_1 \|x_1\|^2 + c v_1$, $Q_{-\{n\}^c} = c^2 v_n \|x_n\|^2 + c v_n$, indicating that quadratic approximation can capture the joint effect amplified by the leverage score by emphasizing the *norm*.

**Submodular property.** Given the challenges of finding an exact solution, it is tempting to explore approximate solutions to MISS with *provable* guarantees. A classical result of Nemhauser et al. [1978] states that so long as the (set) value function satisfies the submodular property, the greedy algorithm will return a solution within a factor $1 - 1/e$ of the optimum. While the value function associated with the first-order approximation is submodular due to linearity, we show in Appendix D.2 that this is generally not the case for $Q_{-S}$. Since the second-order approximation is a more accurate estimation of the actual effect, this suggests that the actual effect is unlikely to be submodular either. Therefore, MISS is expected to be hard even when we allow approximate solutions.

**The role of target function.** Our negative results critically rely on the choice of $x_{\text{test}}$, underscoring the importance of the target function — an issue that has been overlooked in prior research. In addition, we have identified a few target functions in which the influence-based greedy heuristics fail to provide meaningful results: 1) the change of norm, $\phi_1(\theta) = \|\theta - \hat{\theta}\|^2$; 2) the training loss, $\phi_2(\theta) = \|X\theta - y\|^2$. In both of these cases, we have $\nabla_\theta \phi(\hat{\theta}) = 0$, implying that the scores assigned to each sample will also be 0.

**Implication on Linear Datamodeling Score.** Recently, Linear Datamodeling Score (LDS) [Park et al., 2023] has emerged as a prominent metric for evaluating data attribution algorithms [Zheng et al., 2024, Bae et al., 2024]. Central to its design is the assumption that group influence is additive, which we critically examine in our work and reach a negative conclusion. This raises an important question: does a higher LDS result from a truly better data attribution algorithm, or are certain algorithms simply more aligned with the potentially flawed additive assumption? While LDS offers valuable insights into data attribution, we believe it is crucial for the research community to develop evaluation metrics that better capture the *non-additive* and *contextual* nature of training data influence.

**Limitation and future direction.** Despite thorough theoretical and empirical analyses, our study does not offer algorithmic improvements over existing research. We believe solving general MISS is a challenging problem, and hypothesize that there is an inherent trade-off between performance and computational efficiency, in which an increase in performance necessitates additional computing. This pattern is already reflected in the comparison between the vanilla and adaptive greedy algorithms, a trend that will likely continue in future research. To address this challenge, we suggest incorporating the knowledge of target function and data characteristics into algorithmic designs.

# 7   Related work

**(Most) influential subset.**   Since the seminal work of Koh and Liang [2017], which utilized the influence function to identify influential individuals, subsequent research has explored finding an influential *set* of samples [Khanna et al., 2019, Basu et al., 2020, Broderick et al., 2020]. Among them, a notable example is the ZAMinfluence algorithm by Broderick et al. [2020], which builds on the group influence function [Koh et al., 2019] and greedily selects an approximately most influential subset. ZAMinfluence is particularly renowned for its broad applicability: it can be used to improve various dimensions of machine learning such as pre-training [Wang et al., 2023], dataset pruning [Yang et al., 2023b], and trustworthiness [Wang et al., 2022, Sattigeri et al., 2022, Chhabra et al., 2024], as well as to assess the sensitivity of inferential results in multiple domains such as applied econometrics [Attanasio et al., 2015, Angelucci et al., 2015], economics [Finger and Möhring, 2022, Martinez, 2022], and social science [Eubank and Fresh, 2022]. Additionally, Kuschnig et al. [2021] proposed a refined version of ZAMinfluence based on iteratively refitting the model and removing the most influential sample, an approach which was also explored in Yang et al. [2023a].

**Theoretical understanding of MISS.**   Despite its empirical success, the theoretical understanding of ZAMinfluence and other influence-based greedy heuristics remains limited. Giordano et al. [2019a,b] provided finite sample error bounds between the approximated and actual effects, but consistency (i.e., the error uniformly converges to $0$ for all subsets as the sample size goes to infinity) is only achieved as the fraction of removed samples $\alpha$ approaches zero. Fisher et al. [2023] extended the analysis to any fixed $0 < \alpha < 1$, but their consistency is not directly related to the actual effect, thus offering limited insights for MISS. Moitra and Rohatgi [2023], Freund and Hopkins [2023] examined finite-sample stability (i.e., the minimum number of samples that need to be dropped in order to flip the sign of a coordinate) in linear regression and proposed algorithms with provable guarantees, yet they are confined to highly specific scenarios, such as very low dimensions or binary design matrices. Saunshi et al. [2023] explored the additivity assumption in group influence within a different yet less interpretable framework. We position our work as the first to provide a fine-grained analysis of the common practices in MISS, shedding light on the limitations of influence-based greedy heuristics as well as the potential of the adaptive greedy algorithm.

**Multiple outlier detection.**   Classical tools in statistics, such as Cook's distance and its variants, can detect a single outlier in linear regression [Cook, 1986, Chatterjee and Hadi, 1986] and generalized linear models [Wojnowicz et al., 2016]. Nevertheless, they struggle with multiple outliers due to the well-known phenomena of *swamping* and *masking* [Rousseeuw and Leroy, 1987, Hadi and Simonoff, 1993]. This challenge has motivated a line of research in regression diagnostics [Fox, 2019], known as *multiple outlier detection*. Prominent approaches include clustering [Gray and Ling, 1984, Hadi, 1985], influence matrix [Peña and Yohai, 1995], and a class of iterative procedures [Belsley et al., 1980, Hadi and Simonoff, 1993, She and Owen, 2011, Roberts et al., 2015] that resemble Kuschnig et al. [2021]. While seemingly alike, its key distinction from influential subset selection is that the 'outlier' is defined context-independently, rather than with respect to a specific quantity of interest.

**Broader context.**   Our work falls under a broader research area that aims to attribute and interpret model behavior through the lens of data (a.k.a. data attribution). Beyond the influence function, which is central to our study, other popular approaches include the representer point method [Yeh et al., 2018], the data Shapley [Ghorbani and Zou, 2019, Jia et al., 2019], the TracIn algorithm [Pruthi et al., 2020], and more recently, the datamodels [Ilyas et al., 2022]. For a comprehensive review of this subject, we refer readers to Hammoudeh and Lowd [2024]. Finally, we emphasize that MISS should not be confused with data selection [John and Draper, 1975, Kolossov et al., 2023]. While many data attribution algorithms can indeed be applied for data selection (e.g., a recent study Wang et al. [2024] demonstrated that the effectiveness of data Shapley in data selection hinges on the utility function), data selection remains an independent research area. It typically involves *subsampling* a small fraction of the training data to enable effective and *data-efficient* learning or estimation, differing from MISS in its objectives, methodologies, and applications.

# 8   Conclusion

We have provided a comprehensive study of common practices in MISS, revealing the failure modes of influence-based greedy heuristics and uncovering the benefits of adaptivity. We hope our work will enhance the transparency and interpretability of machine learning models by illuminating the collective influence of training data, and serve as a foundation for future algorithmic advancements.

## Acknowledgement

YH and HZ are partially supported by an NSF IIS grant No. 2416897. YH would like to thank Fan Wu for her generous help in the experiments. HZ would like to thank the support from a Google Research Scholar Award. The views and conclusions expressed in this paper are solely those of the authors and do not necessarily reflect the official policies or positions of the supporting companies and government agencies.

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

# A Omitted details from Section 3

## A.1 Preparation work

We start by calculating the OLS estimator, the negative residuals $r_i$'s, the influence estimates $v_i$'s, and the individual effects $A_{-\{i\}}$'s. Suppose there are $c$ copies of $(x_1, y_1)$ and $(x_n, y_n)$, where $c = 1$ unless otherwise noted. Under the label generation process in Eq.(7), we have

$$\hat{\theta} = N^{-1} \left( N\theta^* - c\varepsilon x_1 - pc\varepsilon x_n \right). \tag{14}$$

Therefore, the negative residuals are

$$r_1 = (1 - ch_{11} - pch_{1n})\varepsilon, \quad r_n = (p - pch_{nn} - ch_{1n})\varepsilon, \tag{15}$$

and

$$r_i = -(ch_{1i} + pch_{in})\varepsilon, \quad 2 \leq i \leq n - 1. \tag{16}$$

For $x_{\text{test}} = \frac{x_1 + px_n}{p+1}$, the influence estimates can be calculated as follows:

$$v_1 = \frac{(h_{11} + ph_{1n})(1 - ch_{11} - pch_{1n})\varepsilon}{p+1}, \quad v_n = \frac{(ph_{nn} + h_{1n})(p - pch_{nn} - ch_{1n})\varepsilon}{p+1}, \tag{17}$$

whereas

$$v_i = -\frac{c(h_{1i} + ph_{in})^2 \varepsilon}{p+1} \leq 0, \quad 2 \leq i \leq n - 1. \tag{18}$$

Finally, we have

$$A_{-\{1\}} = \frac{(h_{11} + ph_{1n})(1 - ch_{11} - pch_{1n})\varepsilon}{(p+1)(1 - h_{11})}, \quad A_{-\{n\}} = \frac{(ph_{nn} + h_{1n})(p - pch_{nn} - ch_{1n})\varepsilon}{(p+1)(1 - h_{nn})}, \tag{19}$$

and

$$A_{-\{i\}} = -\frac{c(h_{1i} + ph_{in})^2 \varepsilon}{(p+1)(1 - h_{ii})} \leq 0, \quad 2 \leq i \leq n - 1. \tag{20}$$

We also discuss a few properties of the hat matrix $H$.

**Lemma A.1.** *The leverage scores satisfy:* $h_{11} < \frac{1}{c}$, $h_{nn} < \frac{1}{c}$.

*Proof.* Note the hat matrix is idempotent, i.e., $H^2 = H$. As a consequence, we have

$$h_{11} = ch_{11}^2 + \sum_{i=2}^{n-1} h_{1i}^2 + ch_{1n}^2. \tag{21}$$

Note $\sum_{i=2}^{n-1} x_i x_i^\top$ is invertible, and that $N^{-1} x_1$ is a non-zero vector. As a consequence, we have

$$\sum_{i=2}^{n-1} h_{1i} x_i = \left( \sum_{i=2}^{n-1} x_i x_i^\top \right) N^{-1} x_1 \neq 0, \tag{22}$$

which further implies that the sequence $\{h_{1i}\}_{i=2}^{n-1}$ cannot be all zero. Therefore, we have $h_{11} < \frac{1}{c}$. The same argument applies to $h_{nn}$. $\square$

**Lemma A.2.** *The following inequalities hold:*

$$h_{1n}^2 < h_{11}h_{nn}, \quad \text{and} \quad (1 - ch_{11})(1 - ch_{nn}) < c^2 h_{1n}^2. \tag{23}$$

*Proof.* Since $N$ is positive definite (PD), $P = \sqrt{N^{-1}}$ is well-defined and is invertible. Note $h_{ij} = x_i^\top N^{-1} x_j = \langle Px_i, Px_j \rangle$. Therefore, $h_{1n}^2 < h_{11}h_{nn}$ is equivalent to

$$\langle Px_1, Px_n \rangle < \|Px_1\| \cdot \|Px_n\|. \tag{24}$$

Since $h_{11} > h_{nn}$, we have $x_1 \neq x_n$, and therefore $x_1 \not\parallel x_n$ since their first coordinates are the same. Therefore, Eq.(24) follows from the Cauchy-Schwarz inequality.

For the second inequality, denote $C^\top = (\sqrt{c}x_1, \sqrt{c}x_n) \in \mathbb{R}^{d \times 2}$. Consider the following matrix:

$$S := \begin{pmatrix} N & C^\top \\ C & I_2 \end{pmatrix}. \tag{25}$$

Since the Schur complement of $I_2$: $S/I_2 = N - C^\top I_2 C = \sum_{i=2}^{n-1} x_i x_i^\top \succ 0$, and that $I_2 \succ 0$, we have $S \succ 0$. This further implies that the Schur complement of $N$ is positive definite, i.e.,

$$S/N = I_2 - CN^{-1}C^\top = \begin{pmatrix} 1 - ch_{11} & -ch_{1n} \\ -ch_{1n} & 1 - ch_{nn} \end{pmatrix} \succ 0. \tag{26}$$

As a consequence, we have $\det(S/N) = (1 - ch_{11})(1 - ch_{nn}) - c^2 h_{1n}^2 > 0$. $\qquad \square$

## A.2 Proof of Theorem 3.1

*Proof of Theorem 3.1.* We will show that there exists some $p$, such that

$$1 < \frac{v_n}{v_1} < \frac{1 - h_{nn}}{1 - h_{11}}, \tag{27}$$

and that $v_1$ and $v_n$ are positive. Since $v_i \leq 0$ for $2 \leq i \leq n - 1$, this implies that ZAMinfluence selects $(x_n, y_n)$ and fails to find the most influential sample $(x_1, y_1)$. We will discuss three cases.

**Case 1:** $h_{1n} = 0$. In this case, both $v_1$ and $v_n$ are positive by Lemma A.1. Furthermore, we have

$$\frac{v_n}{v_1} = \frac{h_{nn}(1 - h_{nn})}{h_{11}(1 - h_{11})} \cdot p^2, \tag{28}$$

which is continuous and takes values in $[0, \infty)$. Hence, there exists a $p > 0$ such that Eq.(27) holds.

**Case 2:** $h_{1n} < 0$. When

$$-\frac{h_{1n}}{h_{nn}} < p < -\frac{h_{11}}{h_{1n}}, \tag{29}$$

both $v_1$ and $v_n$ are positive. Note Eq.(29) forms a valid interval by the first inequality in Lemma A.2. Now consider

$$\frac{v_n}{v_1} = \frac{(ph_{nn} + h_{1n})(p - ph_{nn} - h_{1n})}{(h_{11} + ph_{1n})(1 - h_{11} - ph_{1n})}, \tag{30}$$

which is continuous and approaches 0 as $p \to -\frac{h_{1n}}{h_{nn}}$ and approaches $\infty$ as $p \to -\frac{h_{11}}{h_{1n}}$. Hence, there exists a $p > 0$ such that Eq.(27) holds.

**Case 3:** $h_{1n} > 0$. When

$$\frac{h_{1n}}{1 - h_{nn}} < p < \frac{1 - h_{11}}{h_{1n}}, \tag{31}$$

both $v_1$ and $v_n$ are positive. This forms a valid interval by the second inequality in Lemma A.2. The rest of the analysis can be performed similarly as in Case 2. $\qquad \square$

## A.3 Proof of Proposition 3.2

*Proof of Proposition 3.2.* Applying the Woodbury matrix identity, we have

$$(N - X_S^\top I_k X_S)^{-1} = N^{-1} + N^{-1} X_S^\top (I_k - X_S N^{-1} X_S^\top)^{-1} X_S N^{-1} \tag{32}$$

$$= N^{-1} + N^{-1} \sum_{i \in S} \frac{1}{1 - h_{ii}} x_i x_i^\top N^{-1}. \tag{33}$$

Therefore,

$$\hat{\theta}_{-S} - \hat{\theta} = (N - X_S^\top I_k X_S)^{-1} X_{-S}^\top y_{-S} - N^{-1} X^\top y \tag{34}$$

$$= \left( N^{-1} + N^{-1} X_S^\top (I_k - X_S N^{-1} X_S^\top)^{-1} X_S N^{-1} \right)(X^\top y - X_S^\top y_S) - N^{-1} X^\top y \tag{35}$$

$$= N^{-1} X_S^\top (I_k - X_S N^{-1} X_S^\top)^{-1} \left( X_S N^{-1} X^\top y - y_S \right) \tag{36}$$

$$= N^{-1} X_S^\top \left( I_k - X_S N^{-1} X_S^\top \right)^{-1} (X_S \hat{\theta} - y_S), \tag{37}$$

and the actual effect of removing $S$ is

$$A_{-S} := \phi(\hat{\theta}_{-S}) - \phi(\hat{\theta}) = x_{\text{test}}^\top N^{-1} X_S^\top \left( I_k - X_S N^{-1} X_S^\top \right)^{-1} (X_S \hat{\theta} - y_S). \tag{38}$$

$\square$

## A.4 Correspondence between the Neumann series and the Taylor series

We will demonstrate that there is a one-to-one correspondence between the Neumann series $(I_k - M_S)^{-1}$ and the Taylor series of $\hat{\theta}_{-S}(\delta)$. To see this, consider

$$\frac{\partial \hat{\theta}_{-S}(\delta)}{\partial \delta} = n(X^\top X - n\delta X_S^\top X_S)^{-1} X_S^\top (X_S \hat{\theta}_{-S}(\delta) - y_S). \tag{39}$$

From Petersen et al. [2008], we have

$$\frac{\partial K^{-1}}{\partial \delta} = -K^{-1} \frac{\partial K}{\partial \delta} K^{-1} \tag{40}$$

for any invertible symmetric matrix $K$. By induction, we can show that for any $i \geq 1$,

$$\frac{\partial^i \hat{\theta}_{-S}(\delta)}{\partial \delta^i} = (n^i \cdot i!) \cdot (X^\top X - n\delta X_S^\top X_S)^{-1} X_S^\top$$
$$\left[ X_S (X^\top X - n\delta X_S^\top X_S)^{-1} X_S^\top \right]^{i-1} (X_S \hat{\theta}_{-S}(\delta) - y_S). \tag{41}$$

Therefore, by Taylor expansion, we have

$$\hat{\theta}_{-S} = \hat{\theta} + \sum_{i=1}^{\infty} \frac{1}{i!} \left. \frac{\partial^i \hat{\theta}_{-S}(\delta)}{\partial \delta^i} \right|_{\delta=0} \left( \frac{1}{n} \right)^i \tag{42}$$

$$= \hat{\theta} + N^{-1} X_S^\top \left( \sum_{i=1}^{\infty} (X_S N^{-1} X_S^\top)^{i-1} \right) (X_S \hat{\theta} - y_S) \tag{43}$$

$$= \hat{\theta} + N^{-1} X_S^\top \left( \sum_{i=1}^{\infty} M_S^{i-1} \right) (X_S \hat{\theta} - y_S). \tag{44}$$

Therefore, truncating at the $i$-th element in the Neumann series is equivalent to the $i^{\text{th}}$-order Taylor approximation of $\hat{\theta}_{-S}(\delta)$. In particular, first-order approximation corresponds to the identity matrix, which does not concern the leverage scores at all. Conversely, higher-order approximations entail more accurate information on the leverage scores but come at the cost of computational efficiency.

## A.5 Proof of Proposition 3.4

*Proof of Proposition 3.4.* Denote $\theta_{-\{i\}^c}$ as the optimal model parameters after removing all $c$ copies of $(x_i, y_i)$, and $z = \sum_{j=1}^n x_j y_j$. Using the Sherman-Morrison formula, we have

$$\hat{\theta}_{-\{i\}^c} - \hat{\theta} = (N - cx_i x_i^\top)^{-1}(z - cx_i y_i) - N^{-1} z \tag{45}$$

$$= \left( N^{-1} + \frac{cN^{-1} x_i x_i^\top N^{-1}}{1 - ch_{ii}} \right) (z - cx_i y_i) - N^{-1} z \tag{46}$$

$$= \frac{cN^{-1} x_i x_i^\top \hat{\theta}}{1 - ch_{ii}} - cN^{-1} x_i y_i - cN^{-1} x_i y_i \frac{ch_{ii}}{1 - ch_{ii}} \tag{47}$$

$$= \frac{cN^{-1} x_i r_i}{1 - ch_{ii}}. \tag{48}$$

Consequently,

$$A_{-\{i\}^c} = \frac{cx_{\text{test}}^\top N^{-1} x_i r_i}{1 - ch_{ii}}. \tag{49}$$

On the other hand, the influence of removing a single copy is

$$A_{-\{i\}} = \frac{x_{\text{test}}^\top N^{-1} x_i r_i}{1 - h_{ii}}. \tag{50}$$

Therefore,

$$\frac{A_{-\{i\}^c}}{A_{-\{i\}}} = \frac{c \cdot (1 - h_{ii})}{1 - ch_{ii}} > c. \tag{51}$$

□

## A.6   Proof of Theorem 3.5

*Proof of Theorem 3.5.* It suffices to show that there exists some $p$, such that $A_{-\{1\}} < A_{-\{n\}}$ and $A_{-\{1\}^c} > A_{-\{n\}^c}$. This further implies that the failure of LAGS. From Proposition 3.4, it suffices to show there exists some $p$, such that

$$1 < \frac{A_{-\{n\}}}{A_{-\{1\}}} < \frac{(1 - ch_{nn})(1 - h_{11})}{(1 - ch_{11})(1 - h_{nn})}. \tag{52}$$

Note this is a valid interval since

$$(1 - ch_{11})(1 - h_{nn}) = 1 - ch_{11} - h_{nn} + ch_{11}h_{nn} \tag{53}$$
$$< 1 - ch_{nn} - h_{11} + ch_{11}h_{nn} \tag{54}$$
$$= (1 - ch_{nn})(1 - h_{11}), \tag{55}$$

where we use $c \geq 2$ and $h_{11} > h_{nn}$ in the second inequality. Furthermore, Eq.(52) is equivalent to

$$\frac{1 - h_{nn}}{1 - h_{11}} < \frac{v_n}{v_1} < \frac{1 - ch_{nn}}{1 - ch_{11}}, \tag{56}$$

where we use $A_{-\{i\}} = \frac{v_i}{1 - h_{ii}}$. Therefore, we can repeat the analysis in the proof of Theorem 3.1 and conclude the existence of a desired $p$.   □

## A.7   Proof of Theorem 3.6

*Proof of Theorem 3.6.* Recall from Eq.(11) we have

$$A_{-\{1,n\}} = \frac{(1 - h_{11})(1 - h_{nn})(A_{-\{1\}} + A_{-\{n\}}) + h_{1n}x_{\text{test}}^\top N^{-1}(x_1 r_n + x_n r_1)}{(1 - h_{11})(1 - h_{nn}) - h_{1n}^2}. \tag{57}$$

Therefore, $A_{-\{1,n\}} < A_{-\{n\}}$ is equivalent to

$$(1 - h_{11})(1 - h_{nn})A_{-\{1\}} + h_{1n}^2 A_{-\{n\}} + h_{1n}x_{\text{test}}^\top N^{-1}(x_1 r_n + x_n r_1) < 0. \tag{58}$$

Plugging in the formulas of $A_{-\{1\}}, A_{-\{n\}}, r_1, r_n$, Eq.(58) is equivalent to

$$(1 - h_{nn})(h_{11} + ph_{1n})(1 - h_{11} - ph_{1n}) + h_{1n}^2(ph_{nn} + h_{1n})\left(p - \frac{h_{1n}}{1 - h_{nn}}\right)$$
$$< -h_{1n}(h_{11} + ph_{1n})(p - ph_{nn} - h_{1n}) - h_{1n}(ph_{nn} + h_{1n})(1 - h_{11} - ph_{1n}). \tag{59}$$

Combining like terms, we get

$$(h_{11} + ph_{1n})\left((1 - h_{11})(1 - h_{nn}) - ph_{1n}(1 - h_{nn}) + ph_{1n} - h_{1n}(ph_{nn} + h_{1n})\right)$$
$$< -(ph_{nn} + h_{1n})\left(ph_{1n}^2 - \frac{h_{1n}^3}{1 - h_{nn}} - h_{1n}(h_{11} + ph_{1n}) + h_{1n}\right). \tag{60}$$

This could be simplified to

$$(h_{11} + ph_{1n})\left((1 - h_{11})(1 - h_{nn}) - h_{1n}^2\right) < -h_{1n}(ph_{nn} + h_{1n})\frac{(1 - h_{11})(1 - h_{nn}) - h_{1n}^2}{1 - h_{nn}}. \tag{61}$$

Since $(1 - h_{11})(1 - h_{nn}) - h_{1n}^2 > 0$ by Lemma A.2, the above inequality is equivalent to

$$h_{1n}(ph_{nn} + h_{1n}) + (1 - h_{nn})(h_{11} + ph_{1n}) < 0, \tag{62}$$

or

$$h_{1n}p + h_{11}(1 - h_{nn}) + h_{1n}^2 < 0. \tag{63}$$

Now it suffices to show there exists a $p$, such that $A_{-\{1\}}, A_{-\{n\}} > 0$, and that Eq.(63) holds.

**Case 1:** $h_{1n} < 0$. When

$$-\frac{h_{1n}}{h_{nn}} < p < -\frac{h_{11}}{h_{1n}}, \tag{64}$$

both $A_{-\{1\}}$ and $A_{-\{n\}}$ are positive. Furthermore, we have

$$\lim_{p \to -\frac{h_{11}}{h_{1n}}} h_{1n}p + h_{11}(1 - h_{nn}) + h_{1n}^2 = h_{1n}^2 - h_{11}h_{nn} < 0 \tag{65}$$

from Lemma A.2. This proves the existence of a desired $p$.

**Case 2:** $h_{1n} > 0$. When

$$-\frac{h_{11}}{h_{1n}} < p < -\frac{h_{1n}}{h_{nn}}, \tag{66}$$

both $A_{-\{1\}}$ and $A_{-\{n\}}$ are positive. Similarly, we can pick a $p$ that is sufficiently close to $-\frac{h_{11}}{h_{1n}}$, such that $p \neq -1$ and Eq.(63) holds.

Combining the above two cases finishes the proof as desired. $\qquad \square$

# B  Omitted details from Section 4

## B.1  Preparation work

We start by computing the *updated* OLS estimator, the negative residuals, and the individual effects after removing the sample $(x_n, y_n)$. Denote $N' = \sum_{i=1}^{n-1} x_i x_i^\top$, the updated OLS estimator is

$$\hat{\theta}' = (N')^{-1}(N'\theta^* - \varepsilon x_1). \tag{67}$$

Therefore, the updated negative residuals are $r_1' = (1 - h_{11}')\varepsilon$ and $r_i' = -h_{1i}'\varepsilon$ for $2 \leq i \leq n - 1$. By the Sherman-Morrison formula,

$$(N')^{-1} = N^{-1} + \frac{N^{-1}x_n x_n^\top N^{-1}}{1 - x_n^\top N^{-1} x_n} = N^{-1} + \frac{N^{-1}x_n x_n^\top N^{-1}}{1 - h_{nn}}. \tag{68}$$

Therefore, we have

$$h_{1i}' = h_{1i} + \frac{h_{1n}h_{in}}{1 - h_{nn}}, \quad h_{ii}' = h_{ii} + \frac{h_{in}^2}{1 - h_{nn}}, \quad x_i^\top (N')^{-1} x_n = \frac{h_{in}}{1 - h_{nn}} \tag{69}$$

for $1 \leq i \leq n - 1$. Finally, the adjusted individual effects are

$$A'_{-\{1\}} = \frac{x_{\text{test}}^\top N'^{-1} x_1 r_1'}{1 - h_{11}'} = \frac{h_{11}' + p x_1^\top (N')^{-1} x_n}{p + 1}, \tag{70}$$

and

$$A'_{-\{i\}} = \frac{x_{\text{test}}^\top N'^{-1} x_i r_i'}{1 - h_{ii}'} = -\frac{p h_{1i}' x_i^\top N'^{-1} x_n + h_{1i}'^2}{(p + 1)(1 - h_{ii}')} \tag{71}$$

for $2 \leq i \leq n - 1$.

We will also make use of the following lemma.

**Lemma B.1.** *For $2 \leq i \leq n - 1$, $A_{-\{i,n\}} < A_{-\{n\}}$ is equivalent to*

$$\left(h_{1i}h_{in}(1 - h_{nn}) + h_{in}^2 h_{1n}\right)p + \left(h_{1i}(1 - h_{nn}) + h_{in}h_{1n}\right)^2 > 0. \tag{72}$$

*Proof.* From Eq.(11), we have

$$A_{-\{i,n\}} = \frac{(1 - h_{ii})(1 - h_{nn})(A_{-\{i\}} + A_{-\{n\}}) + h_{in}x_{\text{test}}^\top N^{-1}(x_i r_n + x_n r_i)}{(1 - h_{ii})(1 - h_{nn}) - h_{in}^2}. \tag{73}$$

Therefore, $A_{-\{i,n\}} < A_{-\{n\}}$ is equivalent to

$$(1 - h_{ii})(1 - h_{nn})A_{-\{i\}} + h_{in}^2 A_{-\{n\}} + h_{in}x_{\text{test}}^\top N^{-1}(x_i r_n + x_n r_i) > 0. \tag{74}$$

Plugging in the formulas of $A_{-\{i\}}, A_{-\{n\}}, r_i, r_n$, Eq.(74) is equivalent to

$$-(h_{1i} + ph_{in})^2(1 - h_{nn}) + \frac{(ph_{nn} + h_{1n})(p - ph_{nn} - h_{1n})h_{in}^2}{1 - h_{nn}}$$
$$+ h_{in}(h_{1i} + ph_{in})(p - 2ph_{nn} - 2h_{1n}) > 0. \tag{75}$$

Multiplying both side by $(1 - h_{nn})$, the coefficient of $p^2$ is

$$-h_{in}^2(1 - h_{nn})^2 + h_{in}^2 h_{nn}(1 - h_{nn}) + h_{in}^2(1 - h_{nn})(1 - 2h_{nn}) = 0; \tag{76}$$

the coefficient of $p$ is

$$-2h_{1i}h_{in}(1 - h_{nn})^2 + h_{in}^2 h_{1n}(1 - 2h_{nn}) + (1 - h_{nn})h_{in}\left(h_{1i}(1 - 2h_{nn}) - 2h_{1n}h_{in}\right)$$
$$= -h_{1i}h_{in}(1 - h_{nn}) - h_{in}^2 h_{1n}; \tag{77}$$

and the constant term is

$$-(1 - h_{nn})^2 h_{1i}^2 - h_{1n}^2 h_{in}^2 - 2h_{1i}h_{1n}h_{in}(1 - h_{nn}) = -\left(h_{1i}(1 - h_{nn}) + h_{in}h_{1n}\right)^2. \tag{78}$$

Therefore, Eq.(75) is equivalent to

$$\left(h_{1i}h_{in}(1 - h_{nn}) + h_{in}^2 h_{1n}\right)p + \left(h_{1i}(1 - h_{nn}) + h_{in}h_{1n}\right)^2 > 0. \tag{79}$$

$\square$

## B.2 Proof of Proposition 4.1

*Proof of sign consistency.* For $(x_1, y_1)$, plugging Eq.(69) into Eq.(70), we have

$$A'_{-\{1\}} < 0 \iff \left(h_{11} + \frac{h_{1n}^2}{1 - h_{nn}}\right) + p\left(h_{1n} + \frac{h_{1n}h_{nn}}{1 - h_{nn}}\right) < 0 \tag{80}$$

$$\iff h_{1n}p + h_{11}(1 - h_{nn}) + h_{1n}^2 < 0, \tag{81}$$

which aligns with Eq.(63). Therefore, $A_{-\{1,n\}} < A_{-\{n\}} \iff A'_{-\{1\}} < 0$.

For $(x_i, y_i)$ where $2 \leq i \leq n - 1$, plugging Eq.(69) into Eq.(71), we have

$$A'_{-\{i\}} = -\frac{p\left(h_{1i} + \frac{h_{1n}h_{in}}{1 - h_{nn}}\right)\frac{h_{in}}{1 - h_{nn}} + \left(h_{1i} + \frac{h_{1n}h_{in}}{1 - h_{nn}}\right)^2}{(p + 1)\left(1 - h_{ii} - \frac{h_{in}^2}{1 - h_{nn}}\right)} \tag{82}$$

$$= -\frac{ph_{in}\left(h_{1n}h_{in} + h_{1i}(1 - h_{nn})\right) + \left(h_{1i}(1 - h_{nn}) + h_{1n}h_{in}\right)^2}{(p + 1)(1 - h_{nn})s_i}. \tag{83}$$

This implies

$$A'_{-\{i\}} < 0 \iff \left(h_{1i}h_{in}(1 - h_{nn}) + h_{in}^2 h_{1n}\right)p + \left(h_{1i}(1 - h_{nn}) + h_{in}h_{1n}\right)^2 > 0, \tag{84}$$

which aligns with Eq.(72) in Lemma B.1. Therefore, $A_{-\{i,n\}} < A_{-\{n\}} \iff A'_{-\{i\}} < 0$. $\square$

*Proof of order preservation.* Plugging $A_{-\{i\}}, A_{-\{n\}}$ into Eq.(11), we have

$$A_{-\{i,n\}} = \frac{\begin{aligned}-(1 - h_{nn})(h_{1i} + ph_{in})^2 + (1 - h_{ii})(ph_{nn} + h_{1n})(p - ph_{nn} - h_{1n})\\ + h_{in}(h_{1i} + ph_{in})(p - 2ph_{nn} - 2h_{1n})\end{aligned}}{(1 - h_{ii})(1 - h_{nn}) - h_{in}^2}. \tag{85}$$

Denote $s_i = (1 - h_{ii})(1 - h_{nn}) - h_{in}^2 > 0$. In the numerator, the coefficient of $p^2$ is

$$-(1 - h_{nn})h_{in}^2 + h_{nn}(1 - h_{ii})(1 - h_{nn}) + h_{in}^2(1 - 2h_{nn}) = h_{nn}s_i; \tag{86}$$

the coefficient of $p$ is

$$- 2h_{1i}h_{in}(1 - h_{nn}) + (1 - h_{ii})(1 - h_{nn})h_{1n} - (1 - h_{ii})h_{1n}h_{nn}$$
$$- 2h_{1n}h_{in}^2 + h_{1i}h_{in}(1 - 2h_{nn}) \tag{87}$$

$$= -h_{1i}h_{in} - h_{1n}h_{in}^2 + s_i h_{1n} - h_{1n}h_{nn}(1 - h_{ii}) \tag{88}$$

$$= -\frac{1}{1 - h_{nn}}\left((h_{1i}h_{in} + h_{1n}h_{in}^2)(1 - h_{nn}) + h_{1n}h_{nn}h_{in}^2 + s_i h_{1n}(2h_{nn} - 1)\right) \tag{89}$$

$$= -\frac{1}{1 - h_{nn}}\left(h_{in}\left(h_{1i}(1 - h_{nn}) + h_{1n}h_{in}\right) + s_i h_{1n}(2h_{nn} - 1)\right), \tag{90}$$

and the constant term is

$$- (1 - h_{nn})h_{1i}^2 - h_{1n}^2(1 - h_{ii}) - 2h_{1n}h_{1i}h_{in}$$

$$= -\frac{1}{1 - h_{nn}}\left((1 - h_{nn})^2 h_{1i}^2 + 2h_{1n}h_{1i}h_{in}(1 - h_{nn}) + h_{1n}^2 s_i + h_{1n}^2 h_{in}^2\right) \tag{91}$$

$$= -\frac{1}{1 - h_{nn}}\left(\left(h_{1i}(1 - h_{nn}) + h_{1n}h_{in}\right)^2 + h_{1n}^2 s_i\right). \tag{92}$$

Therefore,

$$A_{-\{i,n\}} = \left(h_{nn}p^2 + \frac{(1 - 2h_{nn})h_{1n}}{1 - h_{nn}}p - \frac{h_{1n}^2}{1 - h_{nn}}\right) + B_i, \tag{93}$$

where

$$B_i = -\frac{ph_{in}\left(h_{1i}(1 - h_{nn}) + h_{1n}h_{in}\right) + \left(h_{1i}(1 - h_{nn}) + h_{1n}h_{in}\right)^2}{s_i}. \tag{94}$$

Since $h_{1n}, h_{nn}, p$ are constants, $\{A_{-\{i,n\}}\}_{i=1}^{n-1}$ and $\{B_i\}_{i=1}^{n-1}$ are order-isomorphic. Furthermore, from Eq. (83) we have

$$A'_{-\{i\}} = \frac{B_i}{(p + 1)(1 - h_{nn})}. \tag{95}$$

Therefore, $\{A'_{-\{i\}}\}_{i=2}^{n-1}$ and $\{B_i\}_{i=2}^{n-1}$ are also order-isomorphic. The conclusion then follows from the transitivity of order-isomorphism. $\square$

## B.3 Proof of a technical lemma

We will show that when $A_{-\{1\}}, A_{-\{n\}} > 0$, $A_{-\{1,n\}} < A_{-\{n\}}$ implies $A_{-\{1\}} < A_{-\{n\}}$. This guarantees $(x_n, y_n)$ to be the most influential sample since $A_{-\{i\}} \leq 0$ for $2 \leq i \leq n - 1$.

*Proof.* Plugging in the formulas of $A_{-\{1\}}, A_{-\{n\}}$, we have

$$A_{-\{1\}} < A_{-\{n\}} \iff \left(p - \frac{h_{1n}}{1 - h_{nn}}\right)(ph_{nn} + h_{1n}) > (ph_{1n} + h_{11})\left(1 - \frac{ph_{1n}}{1 - h_{11}}\right). \tag{96}$$

This is equivalent to

$$\left(h_{nn} + \frac{h_{1n}^2}{1 - h_{11}}\right)p^2 + \frac{h_{1n}(h_{11} - h_{nn})}{(1 - h_{11})(1 - h_{nn})}p - \left(h_{11} + \frac{h_{1n}^2}{1 - h_{nn}}\right) > 0. \tag{97}$$

Recall from Eq. (63) that $A_{-\{1,n\}} < A_{-\{n\}}$ is equivalent to $h_{1n}p + h_{11}(1 - h_{nn}) + h_{1n}^2 < 0$. It follows that

$$\frac{h_{1n}(h_{11} - h_{nn})}{(1 - h_{11})(1 - h_{nn})}p - \left(h_{11} + \frac{h_{1n}^2}{1 - h_{nn}}\right) > \frac{h_{1n}(h_{11} - h_{nn})}{(1 - h_{11})(1 - h_{nn})}p + \frac{h_{1n}(1 - h_{11})}{(1 - h_{11})(1 - h_{nn})}p$$
$$\tag{98}$$

$$= \frac{h_{1n}}{1 - h_{11}}p. \tag{99}$$

Therefore, it suffices to show

$$\left(h_{nn} + \frac{h_{1n}^2}{1 - h_{11}}\right) p^2 + \frac{h_{1n}}{1 - h_{11}} p > 0. \tag{100}$$

We now discuss two cases.

**Case 1:** $h_{1n} < 0$. In this case, we must have $p > 0$ to ensure Eq. (63). Therefore, Eq. (100) is equivalent to

$$h_{1n} + \left(h_{nn}(1 - h_{11}) + h_{1n}^2\right) p > 0. \tag{101}$$

Plugging in $p = -\frac{h_{11}(1 - h_{nn}) + h_{1n}^2}{h_{1n}}$, it suffices to show

$$\left(h_{11}(1 - h_{nn}) + h_{1n}^2\right)\left(h_{nn}(1 - h_{11}) + h_{1n}^2\right) > h_{1n}^2. \tag{102}$$

This is true since

$$\left(h_{11}(1 - h_{nn}) + h_{1n}^2\right)\left(h_{nn}(1 - h_{11}) + h_{1n}^2\right)$$
$$= h_{11}h_{nn}(1 - h_{11} - h_{nn}) + h_{1n}^2(h_{11} + h_{nn}) + (h_{11}h_{nn} - h_{1n}^2)^2 \tag{103}$$
$$> h_{1n}^2(1 - h_{11} - h_{nn}) + h_{1n}^2(h_{11} + h_{nn}) = h_{1n}^2. \tag{104}$$

**Case 2:** $h_{1n} > 0$. In this case, we must have $p < 0$ to ensure Eq. (63). Therefore, Eq. (100) is equivalent to

$$h_{1n} + \left(h_{nn}(1 - h_{11}) + h_{1n}^2\right) p < 0. \tag{105}$$

Plugging in $p = -\frac{h_{11}(1 - h_{nn}) + h_{1n}^2}{h_{1n}}$, it suffices to show

$$\left(h_{11}(1 - h_{nn}) + h_{1n}^2\right)\left(h_{nn}(1 - h_{11}) + h_{1n}^2\right) > h_{1n}^2, \tag{106}$$

which is essentially Eq. (102).

Combining the above two cases finishes the proof as desired. $\qquad\square$

# C Omitted details from Section 5

## C.1 Empirical justification with synthetic dataset

We first demonstrate our theory of linear regression empirically, Theorem 4.2 in particular, with a carefully designed synthetic dataset to create the cancellation phenomenon. Firstly, we random sample $\theta^* \in \mathbb{R}^d$ and $X \in \mathbb{R}^{(n-2\cdot c)\times d}$ where each entrance is between $[-1, 1]$. Here, $c$ is the size of two *clusters* that will happen to create the cancellation phenomenon. We then artificially attached an all-one matrix $\mathbb{1} \in \mathbb{R}^{(2\cdot c)\times d}$ to (the bottom of) $X$, which corresponds to the *farmost* features of those two clusters. Then, we create the response $y \in \mathbb{R}^n$ by first calculating the *perfect response* $y^* := X\theta^*$, and perturb it by adding and subtracting some noise $\epsilon$ from the two clusters, respectively. In particular, for each $i \in [2 \cdot c + 1, n]$, we sample a noise $\epsilon_i \sim y_i^* Z$ proportional to its original magnitude $y_i^*$, where $Z \sim \mathcal{N}(1, \sigma^2)$ for some variance $\sigma^2 > 0$. Finally, we note that we create each test data point $x_{\text{test}} \in \mathbb{R}^d$ by again sampling each entry uniformly from $[-1, 1]$.

Intuitively, this training dataset contains two clusters on the opposite side of the ground truth $\theta^*$, hence creating the cancellation phenomenon. For demonstration, we choose $d = 10$, $\sigma^2 = 0.2$, and $n = 1000$ with a cluster size of $c = 50$. The results are reported in Figure 5. We see that when $k < c$, the vanilla greedy and the adaptive greedy algorithm perform similarly. However, when $k > c$, we immediately see a clear separation in terms of the performance of the vanilla greedy and the adaptive greedy algorithm, which gives strong evidence that the adaptive greedy can capture the marginal effect after removing the entire cluster.

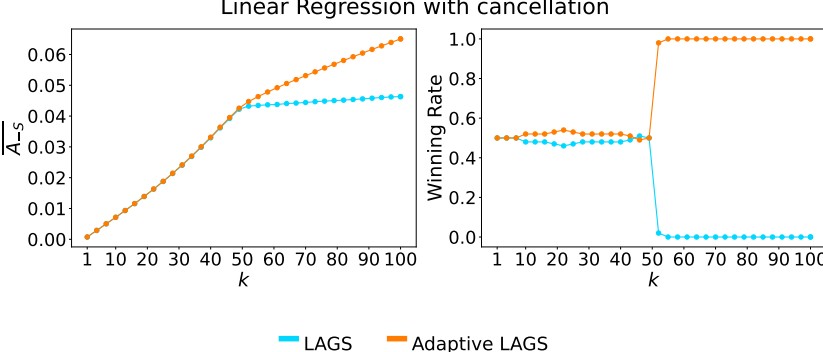

Figure 5: Adaptive Greedy v.s. Greedy Algorithm. **Left**: Averaged actual effect $\overline{A_{-S}}$ measures the averaged actual effect induced by the greedy and adaptive greedy algorithms. **Right**: Winning rate indicates the proportion of instances where one algorithm outperforms the other.

## C.2 Details of the datasets

We detail two of the UCI datasets we chose in our experiments.

- Concrete Compressive Strength [Yeh, 2007]: The dataset contains $1030$ instances and $8$ features.
- Waveform Database Generator [Breiman and Stone, 1988]: It contains $5000$ instances and $21$ features, with three different classes. Since we consider binary classification for logistic regression, we select the first two classes for our experiments, which contain in total $3254$ instances.

The two UCI datasets are licensed under CC-BY 4.0, while the MNIST dataset holds a CC BY-SA 3.0 license.

**Train/valid/test split.** For the first two UCI datasets, we randomly sample $50$ data points as the test set and use the remaining for training. For MNIST, to control the scale of the experiments, we sample $5000$ data points from the train split for training and $50$ data points from the test split for testing.

## C.3 Details of the MLP training

We consider a simple 2-layer MLP with input size $784$ (to match the input size of images from MNIST [LeCun et al., 1998]) and a hidden-size of $128$, with ReLU [Agarap, 2018] as our activation function. We train the model using Stochastic Gradient Descent (SGD) [Ruder, 2016] till convergence, with a learning rate of $0.01$ and momentum of $0.9$. Empirically, we observe that after $30$ epochs the model converges, hence for simplicity, we set the default epochs to be $30$.

**Hyper-parameter selection.** The reported hyper-parameters above were selected via grid search. We swept across hidden unit number (denoted as "width") $\in \{64, 128\}$, learning rate (denoted as "lr") $\in \{0.01, 0.05, 0.1, 0.5\}$, momentum (denoted as $\beta$) $\in \{0.9, 0.95\}$, and training epochs (denoted as "epochs") $\in \{30, 50\}$. For each combination of hyper-parameters, we performed 5-fold cross-validation. We present the comparisons in Table 1, which supported our final choice of the hyper-parameters in the main experiments (width $= 128$, lr $= 0.01$, $\beta = 0.9$, epochs $= 30$).

## C.4 Enhancing computational efficiency for the MLP experiments

As mentioned in Section 5, the adaptive greedy algorithm is time-consuming as every run of the algorithm requires retraining for ($k \times$ number of ensembles) times if only one point is selected at each step. In our case, one evaluation requires around $10^4$ many retraining. Hence, we adopt several efficient approximations to mitigate the computational burden.

Firstly, when computing the vanilla individual influence of training data points for a converged MLP, we leverage one of the most memory and time-efficient approximation algorithms known in the literature named EK-FAC [George et al., 2018] to expedite computation. EK-FAC is efficient enough to deal with large language models, which suffices for our purpose. Additionally, we devise the following two strategies to reduce the computational cost when being adaptive:

Table 1: **Cross-validation performance** for MLP Model on MNIST. Width stands for the width of the hidden layer of the MLP, lr stands for the learning rate, and $\beta$ stands for the momentum.

| width | lr | $\beta$ | epochs | Accuracy | width | lr | $\beta$ | epochs | Accuracy |
|---|---|---|---|---|---|---|---|---|---|
| 64 | 0.01 | 0.9 | 30 | 91.96% | 64 | 0.1 | 0.9 | 30 | 39.36% |
| 128 | 0.01 | 0.9 | 30 | **93.44%** | 128 | 0.1 | 0.9 | 30 | 40.08% |
| 64 | 0.01 | 0.9 | 50 | 92.88% | 64 | 0.1 | 0.9 | 50 | 41.64% |
| 128 | 0.01 | 0.9 | 50 | 93.40% | 128 | 0.1 | 0.9 | 50 | 45.36% |
| 64 | 0.01 | 0.95 | 30 | 92.48% | 64 | 0.1 | 0.95 | 30 | 13.48% |
| 128 | 0.01 | 0.95 | 30 | 93.12% | 128 | 0.1 | 0.95 | 30 | 13.36% |
| 64 | 0.01 | 0.95 | 50 | **93.48%** | 64 | 0.1 | 0.95 | 50 | 10.8% |
| 128 | 0.01 | 0.95 | 50 | **94.68%** | 128 | 0.1 | 0.95 | 50 | 15.71% |
| 64 | 0.05 | 0.9 | 30 | 88.44% | 64 | 0.5 | 0.9 | 30 | 11.68% |
| 128 | 0.05 | 0.9 | 30 | 87.64% | 128 | 0.5 | 0.9 | 30 | 11.68% |
| 64 | 0.05 | 0.9 | 50 | 86.64% | 64 | 0.5 | 0.9 | 50 | 11.68% |
| 128 | 0.05 | 0.9 | 50 | 89.60% | 128 | 0.5 | 0.9 | 50 | 11.68% |
| 64 | 0.05 | 0.95 | 30 | 45.80% | 64 | 0.5 | 0.95 | 30 | 11.04% |
| 128 | 0.05 | 0.95 | 30 | 41.24% | 128 | 0.5 | 0.95 | 30 | 11.20% |
| 64 | 0.05 | 0.95 | 50 | 53.32% | 64 | 0.5 | 0.95 | 50 | 11.20% |
| 128 | 0.05 | 0.95 | 50 | 54.60% | 128 | 0.5 | 0.95 | 50 | 11.20% |

- **Adaptation with steps**: We enhance the adaptive greedy with a tunable parameter, step size $\ell$, i.e., we select the top $\ell$ most influential training points into a tentative most influential subset $S$ at each selection step. The standard adaptive greedy has $\ell = 1$. In our experiment, we set $\ell = 5$ in particular.

- **Warm start**: At each step, we need to obtain a new model that is supposed to be trained without $S$. To make the adaptive greedy algorithm more efficient, we obtain a new model by first initializing the model parameters from the *previous step* (for each seed of the ensemble, respectively), and train without $S$ until convergence. Empirically, we observed that compared to the cold start (which requires 30 epochs to converge), the warm start only requires 8 epochs to converge, significantly reducing the computational time.

## C.5 MLP experiments with multiple random seeds

We repeat the MLP experiments using multiple random seeds and report the results in Figure 4. The randomness in the experiments arises from neural network training. In summary, our results are generally consistent and robust across different random seeds. Specifically, the adaptive greedy algorithm consistently outperforms the vanilla greedy algorithm, though there are some fluctuations in the winning rate.

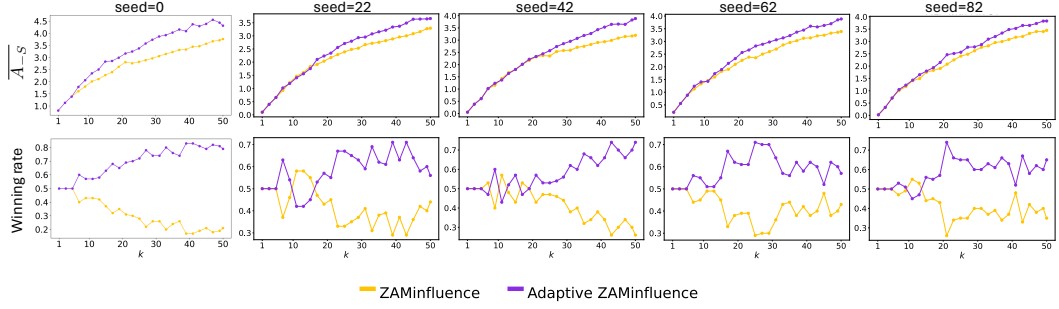

Figure 6: The MLP experiment under different random seeds (0, 22, 42, 62, 82). We report the actual effect and the winning rate. Results in the main paper in Figure 4 were obtained on seed 0.

## C.6 Computational resource and complexity

We conduct our experiments on `Intel(R) Xeon(R) Gold 6338 CPU @ 2.00GHz` with `Nvidia A40 GPU`. All experiments except the MLP experiment are efficient due to parallelization and low memory requirements. Specifically, for linear regression, both experiments on synthetic and UCI datasets run under 20 seconds. As for logistic regression, the experiment finishes in 2 minutes.

On the other hand, for the MLP experiments on MNIST, one step of the adaptive greedy selection algorithm for a test data point on 5000 train data points takes roughly 200 seconds with an average GPU memory usage of `40000MiB`. Therefore, we can't afford any parallelization over test points due to the high memory usage. Without parallelization, using the warm start and a step size of $\ell = 5$, the whole evaluation (5000 train data points, 50 test data points, $k = 50$) takes roughly takes 28 hours.

# D  Omitted details from Section 6

## D.1  Discussion on the quadratic optimization

Recall from Eq.(13) that

$$
\begin{aligned}
Q_{-S} &= x_{\text{test}}^\top N^{-1} X_S^\top \left( I_k + X_S N^{-1} X_S^\top \right) (X_S \hat{\theta} - y_S) \\
&= \sum_{i \in S} x_{\text{test}}^\top N^{-1} x_i r_i + \sum_{i \in S} (x_{\text{test}}^\top N^{-1} x_i) x_i^\top \cdot \sum_{i \in S} x_i r_i.
\end{aligned} \tag{107}
$$

Denote $v = (v_1, \cdots, v_n)^\top$ and $B = (b_{ij})$, where $b_{ij} = (x_{\text{test}}^\top N^{-1} x_i) x_i^\top x_j r_j$. Under the second-order approximation, $k$-MISS can be cast as a constrained quadratic optimization problem:

$$
\max_{w \in \{0,1\}^n} \quad w^\top v + w^\top B w \tag{108}
$$
$$
\text{s.t.} \quad \|w\|_0 \le k
$$

## D.2  Discussion on the submodular property

From Eq.(107), we have

$$
Q_{-S} = \sum_{i \in S} v_i + \sum_{i,j \in S} b_{ij}, \tag{109}
$$

Note $Q_{-S}$ is submodular $\iff$ for every $S_1 \subset S_2$ and index $k \notin S_1$,

$$
Q_{-S_1 \cup \{k\}} - Q_{-S_1} \ge Q_{-S_2 \cup \{k\}} - Q_{-S_2}. \tag{110}
$$

Plugging Eq.(109) into Eq.(110), the submodular property requires that

$$
\sum_{i \in S_2 \setminus S_1} (b_{ik} + b_{ki}) \le 0, \tag{111}
$$

which is equivalent to

$$
b_{ij} + b_{ji} \le 0, \quad \forall i, j \in [n]. \tag{112}
$$

Eq.(112) is unlikely to hold especially if $n$ is large, since it requires that the off-diagonal entries of $S_B := B + B^\top$ are all non-positive. For a more rigorous analysis, we focus on the case where the negative residuals $r_i$'s are i.i.d. and symmetrically distributed with respect to the origin. Denote $s_{ij} = \text{sgn}(x_{\text{test}}^\top N^{-1} x_i x_i^\top x_j)$ for $i, j \in [n]$, and the event in Eq.(112) as $\mathcal{E}$. Under this probability model, we have

$$
\Pr(\mathcal{E}) \le \prod_{i \text{ is odd}} \Pr(s_{i(i+1)} r_{i+1} + s_{(i+1)i} r_i \le 0) = \left( \frac{1}{2} \right)^{\lfloor \frac{n}{2} \rfloor}, \tag{113}
$$

which decays exponentially with $n$.

