# OpenReview forum: "Most Influential Subset Selection: Challenges, Promises, and Beyond"
_NeurIPS.cc/2024/Conference — NeurIPS 2024 poster_

### Official Review · Reviewer_Uyvf · 2024-07-03

**Soundness:** 4
**Presentation:** 4
**Contribution:** 2
**Rating:** 7
**Confidence:** 4

**Summary:**

The paper presents a theoretical analysis of why existing greedy additive methods fail to solve the most influential subset selection (MISS) problem, which aims to find the subset of training data with the largest collective influence. Greedy additive methods (first assigning individual influence scores, ordering and taking the top-k group for example) assume linearity of collective influence and fail to account for the non-linearity due to interactions between samples of the group. Building on the analysis, the paper proposes an adaptive greedy algorithm where the individual influence scores are dynamically updated to capture interactions among samples and ensure that samples of the subset have consistent influence (sign + order of influence are preserved). This algorithm is demonstrated using synthetic and real-world (MNIST) experiments.

**Strengths:**

- The paper is very well written.
- The paper contributes a thorough theoretical analysis of failure modes in existing greedy approaches to solve the most influential subset selection (MISS) problem and therefore shows why these approaches often fail to find meaningful subsets.
- The problem of subset influence is interesting and increasingly relevant given the increasing scale of datasets. The paper is therefore not only relevant for the area of data influence but also data-centric AI overall.

**Weaknesses:**

This paper is mainly a theory paper, focusing on the failure modes of influence-based greedy heuristics for most influential subset selection (MISS). The choice of datasets and experiments to run (synthetic data, MNIST on MLPs) fits the scope of the paper. Yet, if I understand correctly, the proposed alternative approach to solve MISS dynamically with the adaptive greedy algorithm is shown for a subset size of 2 and does not scale to larger subset sizes. In Remark 4.3 (line 288), the authors hypothesize that the method would scale to subsets larger than 2 if the found subsets are truly already belonging to the most influential subset. Given the fragility of influence scores (Basu et al., 2021), I doubt that the scenario in the authors' hypothesis is realistic. I am unsure though if I understood the claim in this remark correctly.

**Questions:**

The paper was very clear, and I only have a few questions to gain a better understanding of the potential impact:
- See weaknesses.
- You mention the second-order group influence functions by Basu et al. (2020) in your discussion as an alternative approach to compute group influence. How does the adaptive greedy approach you suggest compare to the second-order approach by Basu et al. (2020) in terms of computational efficiency (when considering group sizes >2, too)? Is it a faster alternative?
- While I acknowledge that this is mainly a theory paper, I would like to understand the potential impact better. What would be potential application scenarios for finding the **most** influential subset?

**Limitations:**

Yes, limitations are discussed in detail in the discussion section.

---

> ### Author Rebuttal · Authors · 2024-08-07
>
> We thank Reviewer Uyvf for taking the time to review our paper and their constructive feedback. Please find below our point-to-point response.
>
>
> **Remark 4.3.**  We note that starting from Section 3.2, we adopt the closed-form of individual influences ($A_{-{i}}$) instead of the influence estimates. The purpose is to separate the two failure modes: the errors incurred by the influence estimates (discussed in Section 3.1 and in prior works such as Basu et al., (2021)), and the non-additive structure of the group influence. Since the errors of influence estimates are not intrinsic to the greedy heuristics as well as their adaptive variants, it is only reasonable for us to use the ground-truth individual effect in order to study their fundamental properties.
>
> Our positive result and the hypothesis in Section 4 also assume access to the ground-truth individual influence. In this context, the fragility of the influence estimates is a separate issue. The critical question is whether the adaptive greedy algorithm can effectively capture the interactions between samples and thereby solve the more general $k$-MISS problem. This is a challenging open problem, and an important first step is to formally define “cancellation” for more than two samples. We will clarify this in the revision, and leave this as future work.
>
>
> **Second-order group influence functions.**  We note that the approach adopted by Basu et al. (2020) is a simplification of the actual second-order influence function, which is part of a more general framework known as the higher-order infinitesimal jackknife [1] in the literature. The actual second-order influence function involves computing a third-order tensor, which is infeasible for deep neural networks (and much more computationally expensive than the adaptive greedy algorithm). Consequently, Basu et al. (2020) ignored the third-order derivative in their calculations, making their computational complexity roughly on the same order as the vanilla influence function and more efficient than the adaptive greedy algorithm. However, there are two main caveats: 1) The third-order term might contain rich information for large-scale neural networks and classification tasks; 2) The subset selection problem is reframed as a discrete quadratic optimization problem, and while it can be solved efficiently via relaxation and projected gradient descent, the additional step makes it challenging to obtain provable guarantees even in simple linear models.
>
> Beyond computational efficiency, we believe these two approaches capture different types of interactions among samples. The second-order group influence function can detect clusters of samples, corresponding to the amplification effect, whereas the adaptive greedy algorithm can identify samples with cancellation effects, as demonstrated in our analysis. We believe a clever combination of these two approaches holds significant potential and leave this as a topic for future research.
>
>
> **Potential application scenarios.** We will take ZAMinfluence [2], one of the most prominent algorithms in MISS, to discuss the broader impact of MISS.
>
> ZAMinfluence was introduced to assess the sensitivity of applied econometric conclusions to the removal of a small fraction of samples. For example, suppose a conclusion involves determining whether a particular coefficient in a linear regression model is positive. If the sign of the coefficient flips after removing a few samples, we might be concerned as the conclusion is excessively sensitive to a small portion of the data. Conversely, if the sign remains the same regardless of which size-$k$ subset is removed, we can be more confident in the robustness of the conclusion.
>
> ZAMinfluence has been applied to many disciplines, including but not limited to applied econometrics, economics, and social sciences (see Appendix A for a detailed summary). For instance, the following sentence is quoted from [3], a paper in economics: *“Therefore, we use an approach proposed by Broderick et al. (2020) to test if sign and significance of our estimates could conceivably be overturned by removing small fractions of the data with the potentially largest influence on size and sign of estimated effects.”* In experimental studies, the process of collecting samples is often not fully random, and the conclusions drawn from these samples might not be robust. In this case, it is necessary to apply MISS to assess the robustness of conclusions and identify potential sources of sampling bias.
>
> In summary, MISS is framed as a machine learning problem, but it shines through its applications that extend far beyond machine learning, enhancing the reliability of analytical conclusions across a wide range of scientific domains.
>
>
> **References**
>
> [1] Giordano, Ryan, Michael I. Jordan, and Tamara Broderick. "A higher-order swiss army infinitesimal jackknife." arXiv preprint arXiv:1907.12116 (2019).
>
> [2] Broderick, Tamara, Ryan Giordano, and Rachael Meager. "An automatic finite-sample robustness metric: when can dropping a little data make a big difference?." arXiv preprint arXiv:2011.14999 (2020).
>
> [3] Finger, Robert, and Niklas Möhring. "The adoption of pesticide-free wheat production and farmers' perceptions of its environmental and health effects." Ecological Economics 198 (2022): 107463.

---

> > ### Comment · Reviewer_Uyvf · 2024-08-08
> >
> > Thank you for the detailed response.
> >
> > I appreciate the author's suggestion to clarify that their presented method addresses the 2-MISS problem and offers a starting point for the solution for the more general k-MISS problem and believe it will be useful for readers to assess the scope of the paper.
> >
> > My main concern was the potential application impact of this work, where the author's response convinced me of practical scenarios. Hence, I raise my score from 6 -> 7.

---

> > > ### Author Response · Authors · 2024-08-08
> > >
> > > Thank you again for reviewing and acknowledging our work!

---

### Official Review · Reviewer_3kwV · 2024-07-12

**Soundness:** 3
**Presentation:** 2
**Contribution:** 1
**Rating:** 2
**Confidence:** 3

**Summary:**

The authors investigate the Most Influential Subset Selection (MISS) problem, which aims to identify a subset of training samples with the greatest collective influence on machine learning model predictions.
They discuss limitations of prevailing approaches in MISS and highlight issues with influence-based greedy heuristics.
The paper proposes a new method to greedily select the k most influential samples, utilizing adaptive iterative updates to the importance weight of remaining samples that has not been selected by the greedy procedure.

**Strengths:**

- The concept of adaptively updating the influence weight per sample is reasonable.
- The notion of influence is sensible.
- The submodularity approach to studying data subset selection is reasonable.
- The authors discuss the limitations of non-adaptive greedy heuristics for MISS.
- The paper includes an in-depth example and visualizations of the shortcomings of regular greedy algorithms for MISS problems, clearly demonstrating the problems.
- The implementation is publicly available.
- The technical appendix appears to follow a sensible proof strategy; however, I haven't had the chance to formally check the appendix (disclaimer).

**Weaknesses:**

- By their nature, being greedy procedures that iteratively re-evaluate the influence of each individual sample, the proposed algorithms are too inefficient to be applicable to even moderately sized datasets or datasets where subset selection matters most. Consequently, the authors mostly consider small datasets in their experiments.
- As reported in D.5, k-MISS-MLP-MNIST took 28 hours, which is quite wasteful.
- The paper does not formally discuss potential problems and limitations with the applicability of k-MISS in the context of non-linear models; therefore, it remains unclear whether the influence function is truly applicable to non-linear models where samples may have unknown non-linear future influence.
- The influence of randomness from stochastic gradients and initializations (of the MLP) is not discussed or modeled.
- The paper does not convincingly demonstrate that the proposed solution has practical relevance. For example, there is no case study that demonstrates the practical usefulness of the reported results.
- The paper does not convincingly demonstrate that the proposed solution advances our theoretical understanding of MISS.
- The work lacks contextualization regarding highly related research areas, including Data Subset-Selection, Active Learning, Data Shapley, Importance Sampling, Landmark Selection, and Core-set Selection.
- The paper does not compare the proposed solution to established baselines and highly related data, such as SELCO [http://proceedings.mlr.press/v139/s21a/s21a.pdf], which employs submodularity-based approximation strategies to jointly estimate coefficients and the most relevant data subset.
- The paper does not recognize highly influential prior work on Data Subset Selection [http://proceedings.mlr.press/v37/wei15.html].
- The dataset selection lacks representativeness.
- The experimental assessment is based on too few datasets (Concrete Compressive Strength, Waveform Database Generator, and MNIST).
- The test set sizes are very small.
- It remains unclear how robust the proposed algorithm can deal with variations in hyperparameters.
- It remains unclear how robust the notion of influence can handle random effects from initialization and gradient batching.
- The experiment does not report any robustness checks, such as sensitivity analysis or cross-validation.
- The experiments do not include a comparison with state-of-the-art methods from data subset selection or Data Shapley, which makes it difficult to assess its relative performance.

**Questions:**

./.

**Limitations:**

- The authors haven't discussed the implications data sample selection from senstive data:
If the data is user-generated, non-private, and sensitive, this procedure might select and flag data records of certain individuals.
- The experiments are limited and do not showcase practical limitations under different circumstances (see above)

---

> ### Author Rebuttal · Authors · 2024-08-07
>
> We thank Reviewer 3kwV for taking the time to review our paper. Before addressing the reviewer’s comments, we would like to clarify some misconceptions.
>
> - **Thesis and contributions.** Our work falls under the category of learning theory, and our thesis is to advance the theoretical understanding of MISS – an important research topic in the field of data attribution. We do not claim any algorithmic contributions; instead, our aim is to build the foundation for future algorithmic advancements through a comprehensive analysis of the pros and cons of the common practices in MISS. The experiments are designed to corroborate and extend our theoretical findings, instead of showcasing that a particular algorithm has beaten the state-of-the-art.
> - **Related work.** As stated in our general response, data selection and MISS are completely different research topics. They differ in objectives, applications and techniques. As for the relationship between MISS and Data Shapley, 1) influence functions and Data Shapley define the influence/contribution of individual samples in different ways; 2) MISS extends influence functions to modeling the influence of a set of samples. It is clear that MISS and Data Shapley are not comparable.
>
> Below is our point-to-point response:
>
> **Efficiency of the adaptive greedy algorithm.**
> - To be clear, we didn’t propose the adaptive greedy algorithm and certainly were not claiming that it is an all-round solution. One of the main findings of the paper is that the adaptive greedy algorithm trades computational efficiency for performance gain, and the experiments are designed to corroborate and extend this finding.
> - Although efficiency is not on our priority list, we have optimized our processes to the best of our available resources, which is detailed in Appendix D.4.
> - We strongly believe that **the 28-hour experimentation is part of the scientific discovery process and it would be unfair to dismiss it as wasteful.**
>
> **Non-linear models.** Influence functions have been applied to (deep) neural networks ever since the seminar work by Koh and Liang [1]. While our theoretical results are restricted to linear models, the experimental results on MLP clearly demonstrate that our findings can generalize to non-linear models.
>
> **Practical relevance.** As stated above, we did not claim any algorithmic contributions, and our thesis is to advance the theoretical understanding of MISS. For use cases of these algorithms, we refer the reviewer to [2], which is the original paper that proposed ZAMinfluence. We have also summarized a few points in our response to Reviewer Uyvf under “Potential application scenarios”.
>
> **Theoretical understanding.** We do not take such an unjustified claim since our entire work is dedicated to the theoretical understanding of MISS.
>
> **Contextualization.** Please refer to our general response.
>
> **Established baseline and influential prior work.** With due respect, these works are irrelevant to our study. The reasons are detailed in the general response and our clarification on the misconceptions.
>
> **Comparison with mentioned “SOTAs”.** This is incompatible with the thesis of our work, plus they are not comparable. Please refer to our clarification on the misconceptions.
>
> **Robustness to hyperparameters.** We clarify that apart from the randomness of training in the MLP experiments, there is no hyperparameter that needs to be tuned in both the vanilla and adaptive greedy algorithm. For the randomness in training MLPs, please refer to “Randomness in experiments” and our general response.
>
> **Implication on sensitive data.** We believe that the implications of this very specific scenario is beyond the scope of a paper on learning theory that focuses on the fundamentals of the general MISS problem.
>
> **Experiments.**
>
> - **Dataset.**
>
>   - UCI and MNIST are standard datasets in the machine learning community, and MNIST is widely used in the study of influence functions such as [1]. If the reviewer has any “representative” datasets in mind, can the reviewer please share with us?
>   - We consider three different datasets in the experiment section, which we believe is sufficient for a paper on learning theory. This is further supported by Reviewer Uyvf: “The choice of datasets and experiments to run (synthetic data, MNIST on MLPs) fits the scope of the paper”.
>
> - **Size of the test set.**  We want to emphasize that the purpose of the test samples in our setting is very different from their usage in standard machine learning. Here, the test samples serve as the *target function*, and the size could be arbitrary. In fact, we focus on a single test sample in Sections 3 and 4 — in this case, MISS measures the alteration of model behavior on this particular sample of interest.
>
> - **Randomness in experiments.**  For MLP training, we have conducted additional experiments and we show that the results are consistent and robust across different random seeds. Please refer to the general response for details.
>
> - **Robustness checks such as sensitivity analysis and cross-validation.**  For linear regression and logistic regression, there is no need for sensitivity analysis or cross-validation as there’s no hyperparameter tuning. For MLP, we added details of cross-validation in Table 1 of the attached PDF in the general response. We also included them in Appendix D.3 in the draft.
>
>
> **References**
>
> [1] Koh, Pang Wei, and Percy Liang. "Understanding black-box predictions via influence functions." International conference on machine learning. PMLR, 2017.
>
> [2] Broderick, Tamara, Ryan Giordano, and Rachael Meager. "An automatic finite-sample robustness metric: when can dropping a little data make a big difference?." arXiv preprint arXiv:2011.14999 (2020).

---

### Official Review · Reviewer_2zFQ · 2024-07-15

**Soundness:** 3
**Presentation:** 4
**Contribution:** 3
**Rating:** 7
**Confidence:** 4

**Summary:**

The paper explores the challenge of understanding the collective influence of subsets of training data on machine learning models, referred to as the Most Influential Subset Selection (MISS) problem. Traditional influence functions, which focus on individual data points, often miss the more complex interactions within subsets. The authors analyze existing influence-based greedy heuristics, revealing their potential failures, particularly in linear regression, due to errors in the influence function and the non-additive nature of collective influence. They propose an adaptive version of these heuristics, which iteratively updates sample scores to better capture interactions among data points. Their experiments on both synthetic and real-world datasets validate the theoretical findings, demonstrating that the adaptive approach can extend to more complex scenarios like classification tasks and non-linear models.

**Strengths:**

- Addresses a timely important problem of how to leverage pointwise influence estimates to remove datapoints (dataset selection is a very important topic imo, and a related work of [1] recently won a best paper at ICLR). Even if some of the results were known in the literature (see weaknesses), the paper provides a clear story and analysis.
- Exceptional exposition for the most part; very lucid and clear narrative, and addresses both high-level and subtler points (for example, section 6 is a nice discussion of natural follow-up questions)

**Weaknesses:**

- My main concern is that the results presented in Section 3 are sort of a "folklore" in the IF literature (see, e.g., [1] and the references therein). The under-estimation of group effects is also analyzed in earlier work [2], for example.



[1] https://arxiv.org/abs/2309.14563
[2] https://arxiv.org/abs/1905.13289

**Questions:**

- I understand that re-training was modified for the MLP experiments due to computational constraints, but I'm very curious how doing the full correct version would further improve the results in Figure 4. In particular, given the nature of NNs to overfit, I would suspect that partial re-training would not be enough to fully "erase" the influences of deleted points.

- Though the paper focuses on MISS, it would be nice to comment a bit on related problems of dataset selection and coreset selection (which is more of an LISS)

- Figures 1 through 3 should have axes labels. Also, would give more intuition of the examples in the original feature space are also shown.

**Limitations:**

Satisfactory

---

> ### Author Rebuttal · Authors · 2024-08-07
>
> We thank Reviewer 2zFQ for taking the time to review our paper and their constructive feedback. Please find below our point-to-point response.
>
> **Comparison to [1,2].**
>
> - We believe the main and the most fundamental difference between our work and [1] is the topics of study — our work focuses on MISS, whereas [1] concerns data selection. As explained in the general response, these two terms differ in objectives, applications, and techniques (while influence functions are used in both our work and [1], they are used for sub-sampling in [1], which is not a part of our work). Consequently, the conclusions in these works are not transferable (i.e., the sub-optimality of IF in data selection does not necessarily imply its failure in MISS).
> - Regarding [2], the underestimation of IF is only a by-product of our analysis in Section 3.1. In fact, we believe the main contribution of our work lies in Section 3.2, where we provide a comprehensive analysis of the non-additive structure of collective influence. The results in Section 3.1 are not particularly surprising given that influence functions are known to be fragile in prior works; we included it for the sake of completeness.
> - Finally, we have provided a detailed discussion of the theoretical research in MISS in Appendix A.
>
> **Re-training.** We conducted an additional experiment of full re-training following the reviewer’s suggestion. The results are demonstrated in Fig. 1 (right) of the attached PDF in the general response. The main takeaway is that switching from warm start to full re-training does not change our conclusion: the adaptive greedy algorithm consistently outperforms the vanilla one. As a side note, for the winning rate metric, the trend of full re-training does not fully match the one with warm start in Fig 4 of the submission. We believe randomness may have played a significant role in this discrepancy, particularly because full re-training involves more randomness compared to starting from fixed checkpoints. We will further investigate this and include the results of multiple trials with confidence intervals in the revision.
>
> **Data selection/coreset selection.**  Please refer to our general response. We will make sure to clarify the differences between data selection and MISS in the revision.
>
> **Examples in the original feature space (Figs 1-3).**  The samples in Figures 1 to 3 are 2d synthetic data that we crafted for demonstration purposes; they do not correspond to examples (e.g., images) in high-dimensional real-world datasets. The coordinates are generated from Eq. (7) plus some Gaussian noise. We hope this clarifies the reviewer’s question.
>
> **References**
>
> [1] Kolossov, Germain, Andrea Montanari, and Pulkit Tandon. "Towards a statistical theory of data selection under weak supervision." The Twelfth International Conference on Learning Representations.
>
> [2] Koh, Pang Wei W., et al. "On the accuracy of influence functions for measuring group effects." Advances in neural information processing systems 32 (2019).

---

### Official Review · Reviewer_btdJ · 2024-07-21

**Soundness:** 3
**Presentation:** 4
**Contribution:** 3
**Rating:** 7
**Confidence:** 3

**Summary:**

The paper explores the problem of most influential subset selection, that is, selecting a subset of training data points whose removal would change a machine learning model the most. The paper develops further on previous works (most notably of Chatterjee and Hadi, and Kuschnig et al) and shows why the greedy heuristics do not work well in practice. Greedy heuristics operate by computing the influence (e.g., following Koh and Liang) and simply selecting the data points with highest individual influence. The paper shows how indirect interactions can lead to under- and over-estimation of influence. The paper then shows why a specific adaptive method works well in practice. The analysis of the paper is limited to OLS. While the paper does not propose a new influential subset selection algorithm, the analysis conducted in the paper adds important theoretical insights that are likely to be helpful to the research community.

**Strengths:**

1. The paper does a very good job of setting up the problem and explaining the main results. The formalism is clean and easy to understand and the results are accompanied by intuitive explanations, e.g., in Figure 1.
2. The key limitation of greedy heuristic methods (considering the full training dataset) is explained quite well in Section 4.
3. While the paper does not quite add an algorithmic improvement, the theoretical analysis provided is an important building block in developing our understanding of the underlying mechanisms.

**Weaknesses:**

1. Some of the analysis is limited to simpler models like OLS and takes assumptions like the inevitability of the $N=X^TX$ matrix (which may not be the case in real world datasets). This per se is not a big weakness. However, some discussion here on how we expect these results to behave on real world datasets where $N$ is non-invertible, or on models beyond OLS would be greatly helpful.
2. The result of theorem 4.2 seems a bit limited and too tailored to the adaptive greedy algorithm. Specifically, given the steps described in line 251, (remove most influential sample, retrain the model, repeat), and the proposition 4.1 (order preservation between two points), it is not immediately clear if/how much the Theorem 4.2 would extend to 3-MISS and beyond.
3. The title of the paper seems to be quite general and a bit mismatched with the content. While the content shows the weaknesses of existing influence estimation approaches (that too with a OLS model, in a 2-MISS settings and a very specific adaptive algorithm), reading the title feels as if the paper would address a much broader range of models and algorithms.

**Questions:**

1. Section 3.1: To what extent is the finding in Section 3.1 limited to the specific target function used here, that is, the prediction on $x_{test}$? After all, reading the legend in Figure 1, there is not a huge difference on the target function value between points 1 and 8 (0.120 vs. 0.117). More broadly, do we expect the effect of high leverage points to be this large when we take “less noisy” functions like the sign of the prediction on $x_{test}$ or accuracy on the whole test set?
2. Line 171: About correcting the influence with its corresponding leverage score, how does that process look like precisely?
3. If the reviewer understood correctly, $\bar{A_{-S}}$ in Figure 3 shows the actual influence of the two approaches considered. Why not baseline this comparison with the “ground truth influence”, that is, the influence of the actual most influential data points? Clearly, this will ground truth will be very difficult to compute as $k$ increases, but some understanding over smaller values of $k$ might be very helpful for the reader.
4. The results in Figure 4 are mostly as one would expect. However, in the case of MLP, the gap in $\bar{A_{-S}}$ and winning rate starts shrinking around $k=50$. Any reason why that might be happening? How robust are the results to different choices of training data random seed?

**Limitations:**

The paper is generally quite open about the limitations. However, please see the point 3 in weaknesses about the generality of the title.

---

> ### Author Rebuttal · Authors · 2024-08-07
>
> We thank Reviewer btdJ for taking the time to review our paper and their constructive feedback. Please find below our point-to-point response.
>
> **Non-invertible $N$.**  We assume $N$ to be invertible since influence functions rely on the uniqueness of the optimal solution. When this is violated, we can use $L_2$ regularization, which is a standard practice when applying influence functions to deep learning. Our analysis naturally extends to this case. More broadly, we have extended our analysis from OLS to weighted least squares and kernels. However, since these extensions did not provide additional insights, we have opted to present the analysis in its current form. We will ensure these potential generalizations are discussed in the revision.
>
> **Extension to $k$-MISS.**
> - We believe that our analysis being tailored to the adaptive greedy algorithm is not a weakness, as the purpose of Section 4 is to demonstrate that adaptivity helps in a non-trivial way.
> - We acknowledge that under the cancellation setup, our current results are restricted to 2-MISS. We highlight a few challenges: 1) Conceptually, unlike the amplification setup, where it is straightforward to accommodate an arbitrary number of samples, it is not immediately clear how to define “cancellation” for more than two samples. 2) Technically, proving the success of MISS is extremely difficult, as it requires considering all possible subsets, whose size grows exponentially with $k$. In fact, even proving the results for 2-MISS turns out to be highly non-trivial. We will clarify these challenges in the revision and leave them as future research.
>
>
> **Mismatch between title and content.** We fully understand that the reviewer was expecting a broader coverage of algorithms and a more general analysis beyond linear models. Nevertheless,
> - MISS is a relatively new and underdeveloped field. To the best of our knowledge, all algorithms except the second-order group influence function (which we also discussed in Section 6), are based on influence-based greedy heuristics. In this regard, our work pioneers in MISS by systematically studying this dominant strategy.
> - While our theoretical results are limited to linear models and the cancellation setup only concerns 2-MISS, we have conducted experiments on non-linear models and general $k$’s to extend our theoretical analysis.
>
> Therefore, we believe the title is suitable for our work. However, if the reviewer has more concrete suggestions for the title, we would be very open to discussions.
>
>
> **Target functions.**
> - The numbers in Figure 1 are not the actual influences; they are the influence estimates computed by the influence function. We didn’t optimize these numbers hard as it is clear from the proof of Theorem 3.1 that the ratio between them can be arbitrary.
> - Regardless of the target function, the influence function underestimates the change in parameters by $1/(1-h_{ii})$ in linear regression. This is the main reason why samples with high leverage scores could incur a large effect when removed. Based on this fact, it is not hard to generalize our results to other target functions.
> - Take the sign of prediction as an example, we can slightly modify Figure 1 (by shifting the samples along the y-axis), so that the predictions of the original OLS and OLS without sample 8 are negative, but the prediction of OLS without sample 1 is positive. This means that removing sample 1 changes the sign of the prediction, but it does not have the largest influence estimate and is therefore not selected by the greedy algorithm.
> - We posit that the choice of target function is a very important consideration in influence functions yet received insufficient attention from the literature. We have included a discussion in Section 6 in the hope of inspiring future research.
>
>
> **Procedure of correcting the influence estimate.**  In linear regression, it suffices to divide the influence estimate by $(1-h_{ii})$ to obtain the actual individual influence $A_{-\{i\}}$. Please refer to the paragraph under equation (9).
>
> **Comparison with ground truth influence.** We concur that comparing the influences achieved by vanilla/adaptive greedy algorithms to the ground truth is an interesting question. To this end, we conducted additional experiments on a small-scale synthetic dataset.
>
> More specifically, we focus on a binary classification task with logistic regression, and consider the target function $\phi(\theta) = p(z;\theta)$, where $z$ is the test sample and $p(z; \theta)$ is the softmax probability assigned to the correct class. The synthetic data is generated from a mixture of Gaussians: given $\sigma$, we sample $25$ training data points from $\mathcal{N}(c_i, \sigma I_2)$ for each $i\in \{0,1\}$, where $c_0 = (1, 0)$ and $c_1 = (-1, 0)$. The test sample $z$ is uniformly sampled from one of the Gaussians. We conducted the experiments for various $\sigma$’s, and for each $\sigma$, repeated the experiment 20 times using different random seeds to generate the train/test data. Finally, we report the averaged ratio of the obtained influence over the ground-truth influence.
>
> The results are demonstrated in Fig. 2 of the attached PDF in the general response. In summary, the adaptive greedy algorithm outperforms the vanilla counterpart on all $\sigma$’s, and recovers the ground truth subset when $\sigma$ is large.
>
> **Robustness of results to randomness in training.** We believe the drop around $k=50$ is due to randomness. To verify this hypothesis, we conducted two experiments targeting at different randomness in the process. We refer the reviewer to the general response for the detailed setting and the general takeaways of the experiments. Particularly, the first experiment (randomness in evaluation, Fig. 1 top left) clearly demonstrates that the drop in Fig. 4 of the submission is only due to randomness, as we have fixed the selected subset but the new figure does not exhibit the same drop.

---

> > ### Comment · Reviewer_btdJ · 2024-08-14
> > **Raising my score**
> >
> > Thank you for the detailed response. Most of my concerns were addressed and I am raising my score to accept. It would be great to add the additional discussion here on target functions and robustness to the final version of the paper.

---

### Author Rebuttal · Authors · 2024-08-07

We express our sincere gratitude to the reviewers for their detailed review. It is encouraging to see that the reviewers acknowledged the significance of the topic of study: “Addresses a timely important problem”, and “The paper is therefore not only relevant for the area of data influence but also data-centric AI overall”. It is also great to hear that our writing was described as “Exceptional exposition for the most part”, “very lucid and clear narrative”, “and “does a very good job of setting up the problem and explaining the main results”.

The following comment addresses some common questions raised by multiple reviewers. We will attend to specific feedback from the reviewers in our individual responses.

**Related literature.** (Reviewer 2zFQ, 3kwV)
- We would like to clarify that data selection/coreset selection are distinct research areas compared to most influential subset selection (MISS), despite their similar names.
  - In terms of *objectives*, data selection aims to identify the most informative training examples for effective learning or estimation, whereas MISS aims to identify a set of training samples that will maximize the alteration of model behaviors.
  - In terms of *applications*, data selection mostly concerns data efficiency, whereas MISS is typically used for diagnosis (e.g., whether the inferential results based on machine learning models are robust to small variation in data).
  - In terms of *techniques*, MISS is largely built upon influence functions, whereas data selection is typically centered around sub-sampling.
- Due to these differences, data selection/coreset selection are usually not mentioned in the literature of influence function/influential subset selection. Nevertheless, we are aware of the inspiring work [1] mentioned by Reviewer 2zFQ, and we have already discussed the broader context of MISS in Appendix A, including a brief discussion of data selection. We will further clarify these differences and move the literature review to the main text in the camera-ready version.
- Finally, Data Shapley and influence functions are two different approaches to modeling the influence of individual training samples. Influence function serves as an approximation of leave-one-out (LOO), whereas Data Shapley satisfies the equitable valuation conditions. Both methods have their own merits and are not directly comparable.


**Robustness of the MLP experiment.**  (Reviewer btdJ, 3kwV) We appreciate the reviewers’ suggestion on analyzing the robustness of results to randomness in training. We conducted additional experiments using different random seeds, with the results presented in the attached PDF. Concretely, we examined the randomness in both the evaluation step (Fig. 1 left) and the selection step (Fig. 1 middle). In both cases, randomness arises from neural network training — to select the most influential subset, the adaptive greedy algorithm involves repeatedly retraining the MLP after selecting and excluding a new sample; to evaluate a selected subset, we need to train an MLP on the set of training samples excluding the subset. When examining the randomness in the evaluation step, we held the selected subsets fixed (the same ones used to produce Fig. 4 in the submission) and ran the evaluation using a different random seed. When examining the randomness in the selection step, we ran MISS with a different random seed, and then evaluated the newly obtained subsets. The latter scenario involves more randomness.

In summary, our results are generally consistent and robust across different random seeds. Specifically, the adaptive greedy algorithm consistently outperforms the vanilla greedy algorithm, though there are some fluctuations in the winning rate (e.g., Fig. 1, middle). We believe this is still due to the inherent randomness in model training. However, due to limited time and computational resources, we could only run each experiment once during the rebuttal phase. We will include more results with multiple trials in the revision.

---

### Decision · Program_Chairs · 2024-09-25

**Decision:**

Accept (poster)

**Comment:**

This paper provides a clear contribution to our understanding of most important subset selection (MISS), namely, the failure of the greedy approach in the case of linear regression, and the success of adaptive greedy for 2-MISS.  The paper clearly lays out this thesis and provides good discussion of other works and approaches, such as using submodularity.  The reviewers point out the limited nature of the empirical studies, but when taking the clarity of the exposition into account as well as the strong theoretical results, this paper is a clear contribution.

Pros:
1. Timely paper that is likely to guide future work on MISS
2. Exceptional exposition, which clearly addresses their primary thesis
3. The theoretical analysis is instructive, and builds intuition for solving future problems surrounding MISS

Cons:
1. The limited nature of the empirical study
2. The positive result is limited to 2-MISS and it is unclear if this work is extensible to k-MISS more generally